# Spike-driven Transformer V2: Meta Spiking Neural Network Architecture Inspiring the Design of Next-generation Neuromorphic Chips

**Man Yao**[1,2]**, Jiakui Hu**[3]**, Tianxiang Hu**[1]**, Yifan Xu**[1]**, Zhaokun Zhou**[3,2]**,
Yonghong Tian**[2,3]**, Bo Xu**[1]**, Guoqi Li**[1,4*]

[1]Institute of Automation, Chinese Academy of Sciences, Beijing, China
[2]Peng Cheng Laboratory, Shenzhen, Guangzhou, China
[3]Peking University, Beijing, China
[4]Key Laboratory of Brain Cognition and Brain-inspired Intelligence Technology, Beijing, China

## Abstract

Neuromorphic computing, which exploits Spiking Neural Networks (SNNs) on neuromorphic chips, is a promising energy-efficient alternative to traditional AI. CNN-based SNNs are the current mainstream of neuromorphic computing. By contrast, no neuromorphic chips are designed especially for Transformer-based SNNs, which have just emerged, and their performance is only on par with CNN-based SNNs, offering no distinct advantage. In this work, we propose a general Transformer-based SNN architecture, termed as "Meta-SpikeFormer", whose goals are: i) *Lower-power*, supports the spike-driven paradigm that there is only sparse addition in the network; ii) *Versatility*, handles various vision tasks; iii) *High-performance*, shows overwhelming performance advantages over CNN-based SNNs; iv) *Meta-architecture*, provides inspiration for future next-generation Transformer-based neuromorphic chip designs. Specifically, we extend the Spike-driven Transformer in Yao et al. (2023b) into a meta architecture, and explore the impact of structure, spike-driven self-attention, and skip connection on its performance. On ImageNet-1K, Meta-SpikeFormer achieves 80.0% top-1 accuracy (55M), surpassing the current state-of-the-art (SOTA) SNN baselines (66M) by 3.7%. This is the first direct training SNN backbone that can simultaneously supports classification, detection, and segmentation, obtaining SOTA results in SNNs. Finally, we discuss the inspiration of the meta SNN architecture for neuromorphic chip design. Source code and models are available at `https://github.com/BICLab/Spike-Driven-Transformer-V2`.

## 1 Introduction

The ambition of SNNs is to become a low-power alternative to traditional machine intelligence (Roy et al., 2019; Li et al., 2023). The unique *spike-driven* is key to realizing this magnificent concept, i.e., *only a portion of spiking neurons are ever activated to execute sparse synaptic ACcumulate (AC)* when SNNs are run on neuromorphic chips (Roy et al., 2019). Neuromorphic computing is essentially an algorithm-hardware co-design paradigm (Frenkel et al., 2023). Biological neurons are modeled as spiking neurons and somehow form SNNs at the algorithmic level(Maass, 1997a). Neuromorphic chips are then outfitted with spike-driven SNNs at the hardware level (Schuman et al., 2022).

CNN-based SNNs are currently the common spike-driven design. Thus, typical neuromorphic chips, such as TrueNorth (Merolla et al., 2014), Loihi (Davies et al., 2018), Tianjic (Pei et al., 2019), etc., all support spike-driven Conv and MLP operators. Nearly all CNN-era architectures, e.g., VGG (Simonyan & Zisserman, 2015), ResNet (He et al., 2016b), etc., can be developed into corresponding SNN versions (Wu et al., 2021). As Transformer (Vaswani et al., 2017) in ANNs has shown great

---

*Corresponding author, guoqi.li@ia.ac.cn

potential in various tasks (Dosovitskiy et al., 2021), some Transformer-based designs have emerged in SNNs during the past two years (Zhang et al., 2022b;c; Han et al., 2023; Zhou et al., 2023).

Most Transformer-based SNNs fail to take advantage of the low-power of SNNs because they are not spike-driven. Typically, they retain the energy-hungry Multiply-and-ACcumulate (MAC) operations dominated by vanilla Transformer, such as scaled dot-product (Han et al., 2023), softmax (Leroux et al., 2023), scale (Zhou et al., 2023), etc. Recently, Yao et al. (2023b) developed a spike-driven self-attention operator, integrating the spike-driven into Transformer for the first time. However, while the Spike-driven Transformer Yao et al. (2023b) with only sparse AC achieved SOTA results in SNNs on ImageNet-1K, it has yet to show clear advantages over Conv-based SNNs.

In this work, we advance the SNN domain by proposing Meta-SpikeFormer in terms of *performance* and *versatility*. Since Vision Transformer (ViT) (Dosovitskiy et al., 2021) showed that Transformer can perform superbly in vision, numerous studies have been produced (Han et al., 2022). Recently, Yu et al. (2022a;b) summarized various ViT variants and argued that there is general architecture abstracted from ViTs by not specifying the token mixer (self-attention). Inspired by this work, we investigate the meta architecture design in Transformer-based SNNs, involving three aspects: *network structure*, *skip connection (shortcut)*, *Spike-Driven Self-Attention (SDSA)* with fully AC operations.

We first align the structures of Spike-driven Transformer in Yao et al. (2023b) with the CAFormer in Yu et al. (2022b) at the macro-level. Specifically, as shown in Fig. 2, the original four spiking encoding layers are expanded into four Conv-based SNN blocks. We experimentally verify that early-stage Conv blocks are important for the performance and versatility of SNNs. Then we design Conv-based and Transformer-based SNN blocks at the micro-level (see Table 5). For instance, the generation of spike-form Query ($Q_S$), Key ($K_S$), Value ($V_S$), three new SDSA operators, etc. Furthermore, we test the effects of three shortcuts based on the proposed Meta-SpikeFormer.

We conduct a comprehensive evaluation of Meta-SpikeFormer on four types of vision tasks, including image classification (ImageNet-1K (Deng et al., 2009)), event-based action recognition (HAR-DVS (Wang et al., 2022), currently the *largest* event-based human activity recognition dataset), object detection (COCO (Lin et al., 2014)), and semantic segmentation (ADE20K (Zhou et al., 2017), VOC2012 (Everingham et al., 2010)). The main contributions of this paper are as follows:

- **SNN architecture design.** We design a meta Transformer-based SNN architecture with only sparse addition, including macro-level Conv-based and Transformer-based SNN blocks, as well as some micro-level designs, such as several new spiking convolution methods, the generation of $Q_S$, $K_S$, $V_S$, and three new SDSA operator with different computational complexities, etc.

- **Performance.** The proposed Meta-SpikeFormer enables the performance of the SNN domain on ImageNet-1K to achieve 80% for the first time, which is 3.7% higher than the current SOTA baseline but with 17% fewer parameters (55M vs. 66M).

- **Versatility.** To the best of our knowledge, Meta-SpikeFormer is the first direct training SNN backbone that can handle image classification, object detection, semantic segmentation concurrently. We achieve SOTA results in the SNN domain on all tested datasets.

- **Neuromorphic chip design.** We thoroughly investigate the general components of Transformer-based SNN, including structure, shortcut, SDSA operator. And, Meta-SpikeFormer shows significant performance and versatility advantages over Conv-based SNNs. This will undoubtedly inspire and guide the neuromorphic computing field to develop Transformer-based neuromorphic chips.

## 2 RELATED WORK

**Spiking Neural Networks** can be simply considered as Artificial Neural Networks (ANNs) with bio-inspired spatio-temporal dynamics and spike (0/1) activations (Li et al., 2023). Spike-based communication enables SNNs to be spike-driven, but the conventional backpropagation algorithm (Rumelhart et al., 1986) cannot be applied directly because the spike function is non-differentiable. There are typically two ways to tackle this challenge. One is to discrete the trained ANNs into corresponding SNNs through neuron equivalence (Deng & Gu, 2021; Hu et al., 2023), i.e., ANN2SNN. Another is to train SNNs directly, using surrogate gradients (Wu et al., 2018; Neftci et al., 2019). In this work, we employ the direct training method due to its small timestep and adaptable architecture.

**Backbone in Conv-based SNNs.** The architecture of the Conv-based SNN is guided by residual learning in ResNet (He et al., 2016b;a). It can be roughly divided into three categories. Zheng et al. (2021) directly copied the shortcut in ResNet and proposed a tdBN method, which expanded SNN from several layers to 50 layers. To solve the degradation problem of deeper Res-SNN, Fang et al. (2021) and Hu et al. (2024) proposed SEW-Res-SNN and MS-Res-SNN to raise the SNN depth to more than 100 layers. Then, the classic ANN architecture can have a corresponding SNN direct training version, e.g., attention SNNs (Yao et al., 2021; 2023d;a;c), spiking YOLO (Su et al., 2023), etc. Unfortunately, current CNN-based SNNs fail to demonstrate generality in vision tasks.

**Vision Transformers.** After ViT (Dosovitskiy et al., 2021) showed the promising performance, improvements and discussions on ViT have gradually replaced traditional CNNs as the mainstay. Some typical work includes architecture design (PVT (Wang et al., 2021a), MLP-Mixer (Tolstikhin et al., 2021)), enhancement of self-attention (Swin (Liu et al., 2021), Twins (Chu et al., 2021)), training optimization (DeiT (Touvron et al., 2021), T2T-ViT (Yuan et al., 2021)), efficient ViT (Katharopoulos et al., 2020; Xu et al., 2022), etc. In this work, we aim to reference and explore a meta spiking Transformer architecture from the cumbersome ViT variants available to bridge the gap between SNNs and ANNs, and pave the way for future Transformer-based neuromorphic chip design.

**Neuromorphic Chips.** Neuromorphic hardware is non-von Neumann architecture hardware whose structure and function are inspired by brains Roy et al. (2019). Typical neuromorphic chip features include collocated processing and memory, spike-driven computing, etc (Schuman et al., 2022). Functionally speaking, neuromorphic chips that are now on the market either support solely SNN or support hybrid ANN/SNN (Li et al., 2023). The former group consists of TrueNorth (Merolla et al., 2014), Loihi (Davies et al., 2018), Darwin (Shen et al., 2016), etc. The latter includes the Tianjic series (Pei et al., 2019; Ma et al., 2022), SpiNNaker 2 (Höppner et al., 2021). All of these chips enable CNN-based SNNs, but none of them are designed to support Transformer-based SNNs.

## 3 SPIKE-DRIVEN TRANSFORMER V2: META-SPIKEFORMER

### 3.1 THE CONCEPT OF META TRANSFORMER ARCHITECTURE IN ANNS

The self-attention (serves as a *token mixer*) mechanism for aggregating information between different spatial locations (tokens) has long been attributed to the success of Transformer. With the deepening of research, researchers have found that token mixer can be replaced by spatial Multi-Layer Perception (MLP) (Tolstikhin et al., 2021), Fourier Transform (Guibas et al., 2022), etc. Consequently, Yu et al. (2022a;b) argue that compared with a specific token mixer, a genera meta Transformer block (Fig. 1), is more essential than a specific token mixer for the model to achieve competitive performance.

Specifically, the input is first embedded as a feature sequence (tokens) (Vaswani et al., 2017; Dosovitskiy et al., 2021):

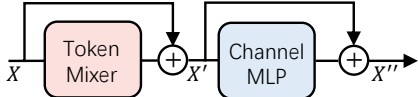

Figure 1: Meta Transformer Block.

$$X = \text{InputEmbedding}(I), \quad (1)$$

where $I \in \mathbb{R}^{3 \times H \times W}$ and $X \in \mathbb{R}^{N \times D}$. 3, $H$, and $W$ denote channel, height and width of the 2D image. $N$ and $D$ represent token number and channel dimension respectively. Then the token sequence $X$ is fed into repeated meta Transformer block, one of which can be expressed as (Fig. 1)

$$X' = X + \text{TokenMixer}(X), \quad (2)$$

$$X'' = X' + \text{ChannelMLP}(X'), \quad (3)$$

where $\text{TokenMixer}(\cdot)$ means token mixer mainly for propagating spatial information among tokens, $\text{ChannelMLP}(\cdot)$ denotes a channel MLP network with two layers. $\text{TokenMixer}(\cdot)$ can be self-attention (Vaswani et al., 2017), spatial MLP (Touvron et al., 2021), convolution (Yu et al., 2022b), pooling (Yu et al., 2022a), linear attention (Katharopoulos et al., 2020), identity map (Wang et al., 2023b), etc, with different computational complexity, parameters and task accuracy.

### 3.2 SPIKING NEURON LAYER

Spiking neuron layer incorporates spatio-temporal information into membrane potentials, then converts them into binary spikes for spike-driven computing in the next layer. We adopt the standard

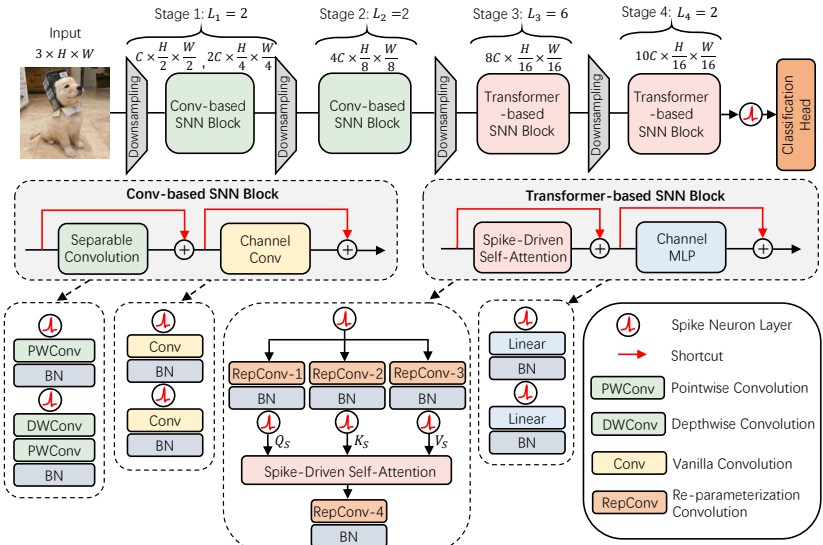

Figure 2: The overview of Meta-SpikeFormer. At the macro level, we refer to the general vision Transformer architecture in Yu et al. (2022a;b) and align Spike-driven Transformer (Yao et al., 2023b) with it. The main macro-level alteration is that we enlarge the spike coding layer from four Conv SNN layers to four Conv-based SNN blocks. At the micro level, we use the meta Transformer block in Fig. 1 as the basis to upgrade to Conv-based and Transformer-based SNN blocks (see Table 5), such as Channel Conv, SDSA operations, etc., to bring them more in line with SNN features.

Leaky Integrate-and-Fire (LIF) (Maass, 1997b) spiking neuron layer, whose dynamics are:

$$U[t] = H[t-1] + X[t], \tag{4}$$

$$S[t] = \mathrm{Hea}\left(U[t] - u_{th}\right), \tag{5}$$

$$H[t] = V_{reset}S[t] + (\beta U[t])\left(\mathbf{1} - S[t]\right), \tag{6}$$

where $X[t]$ ($X[t]$ *can be obtained through spike-driven operators such as Conv, MLP, and self-attention*) is the spatial input current at timestep $t$, $U[t]$ means the membrane potential that integrates $X[t]$ and temporal input $H[t-1]$. $\mathrm{Hea}(\cdot)$ is a Heaviside step function which equals 1 for $x \geq 0$ and 0 otherwise. When $U[t]$ exceeds the firing threshold $u_{th}$, the spiking neuron will fire a spike $S[t]$, and temporal output $H[t]$ is reset to $V_{reset}$. Otherwise, $U[t]$ will decay directly to $H[t]$, where $\beta < 1$ is the decay factor. For simplicity, we mainly focus on Eq. 5 and re-write the spiking neuron layer as $\mathcal{SN}(\cdot)$, with its input as membrane potential tensor $U$ and its output as spike tensor $S$.

### 3.3 META-SPIKEFORMER

In SNNs, the input sequence $I \in \mathbb{R}^{T \times 3 \times H \times W}$, where $T$ denote timestep. For example, images are repeated $T$ times when dealing with a static dataset. To ease of understanding, we subsequently assume $T = 1$ when describing the architectural details of Meta-SpikeFormer.

**Overall Architecture.** Fig. 2 shows the overview of Meta-SpikeFormer, where Conv-based and Transformer-based SNN blocks are both variants of the meta Transformer block in Sec 3.1. In Spike-driven Transformer (Yao et al., 2023b), the authors exploited four Conv layers before Transformer-based blocks for encoding. By contrast, in the architectural form of Conv+ViT in ANNs, there are generally multiple stages of Conv blocks (Han et al., 2022; Xiao et al., 2021). We follow this typical design in ANNs, setting the first two stages to Conv-based SNN blocks, and using a pyramid structure (Wang et al., 2021a) in the last two Transformer-based SNN stages. Note, to control parameter number, we set the channels to $10C$ in stage 4 instead of the typical double ($16C$). Fig. 2 is our recommended architecture. Other alternatives and their impacts are summarized in Table 5.

**Conv-based SNN Block** uses the inverted separable convolution module $\mathrm{SepConv}(\cdot)$ with $7 \times 7$ kernel size in MobileNet V2 (Sandler et al., 2018) as $\mathrm{TokenMixer}(\cdot)$, which is consistent with (Yu et al., 2022b). But, we change $\mathrm{ChannelMLP}(\cdot)$ with $1 \times 1$ kernel size in Eq. 3 to $\mathrm{ChannelConv}(\cdot)$ with

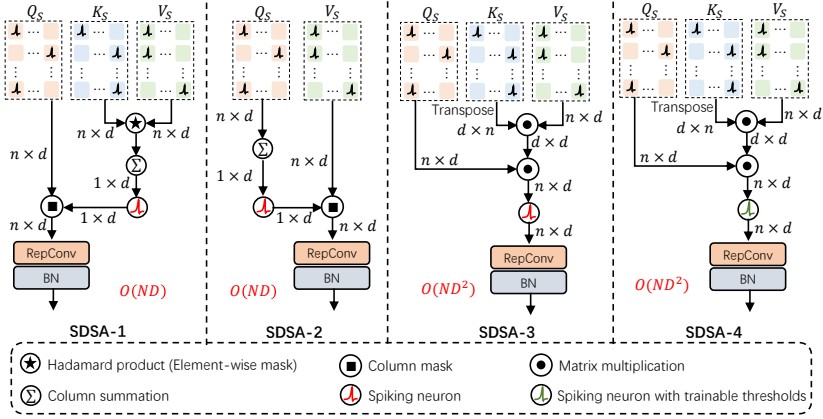

Figure 3: Spike-Driven Self-Attention (SDSA) modules with different computational complexity. SDSA-1 is the operator used in Yao et al. (2023b). SDSA-2/3/4 is the newly designed operator in this paper. We exploit SDSA-3 by default. All SDSAs only have addition, no softmax and scale.

$3 \times 3$ kernel size. The stronger inductive is empirically proved to significantly improve performance (see Table 5). Specifically, the Conv-based SNN block is written as:

$$U' = U + \text{SepConv}(U), \tag{7}$$

$$U'' = U' + \text{ChannelConv}(U'), \tag{8}$$

$$\text{SepConv}(U) = \text{Conv}_{\text{pw2}}(\text{Conv}_{\text{dw}}(\mathcal{SN}(\text{Conv}_{\text{pw1}}(\mathcal{SN}(U))))), \tag{9}$$

$$\text{ChannelConv}(U') = \text{Conv}(\mathcal{SN}(\text{Conv}(\mathcal{SN}(U')))). \tag{10}$$

where $\text{Conv}_{\text{pw1}}(\cdot)$ and $\text{Conv}_{\text{pw2}}(\cdot)$ are pointwise convolutions, $\text{Conv}_{\text{dw}}(\cdot)$ is depthwise convolution (Chollet, 2017), $\text{Conv}(\cdot)$ is the normal convolution. $\mathcal{SN}(\cdot)$ is the spike neuron layer in Sec 3.2.

**Transformer-based SNN Block** contains an SDSA module and a two-layered ChannelMLP$(\cdot)$:

$$Q_S = \mathcal{SN}(\text{RepConv}_1(U)), K_S = \mathcal{SN}(\text{RepConv}_2(U)), V_S = \mathcal{SN}(\text{RepConv}_3(U)), \tag{11}$$

$$U' = U + \text{RepConv}_4(\text{SDSA}(Q_S, K_S, V_S)), \tag{12}$$

$$U'' = U' + \text{ChannelMLP}(U'), \tag{13}$$

$$\text{ChannelMLP}(U') = \mathcal{SN}(\mathcal{SN}(U')W_1)W_2, \tag{14}$$

where $\text{RepConv}(\cdot)$ is the re-parameterization convolutions (Ding et al., 2021) with kernel size $3 \times 3$, $W_1 \in \mathbb{R}^{C \times rC}$ and $W_2 \in \mathbb{R}^{rC \times C}$ are learnable parameters with MLP expansion ratio $r = 4$. Note, both the input $(Q_S, K_S, V_S)$ and output of $\text{SDSA}(\cdot)$ will be reshaped. We omit this for simplicity.

**Spike-Driven Self-Attention (SDSA).** The main difference of SDSA over vanilla self-attention with $O(N^2D)$ in Dosovitskiy et al. (2021) lies in *three* folds: i) Query, Key, Value are spiking tensors; ii) The operations among $Q_S$, $K_S$, $V_S$ do not have softmax and scale; iii) The computational complexity of $\text{SDSA}(\cdot)$ is linear with the token number $N$. Four SDSA operators are given in Fig. 3. SDSA-1 is proposed in Yao et al. (2023b). SDSA-2/3/4 are new operators designed in this work. The main difference between them is the operation between $Q_S$, $K_S$, $V_S$. SDSA-1/2 primarily work with Hadamard product while SDSA-3/4 use matrix multiplication. Spike-driven matrix multiplication can be converted to additions via addressing algorithms (Frenkel et al., 2023). Spike-driven Hadamard product can be seen as a mask (AND logic) operation with almost no cost. Thus, SDSA-1/2/3/4 all only have sparse addition. Details of SDSAs and energy evaluation are given in Appendix A and B.

In this work, we use SDSA-3 by default, which is written as:

$$\text{SDSA}_3(Q_S, K_S, V_S) = \mathcal{SN}_s\left(Q_S\left(K_S^{\text{T}}V_S\right)\right) = \mathcal{SN}_s((Q_SK_S^{\text{T}})V_S). \tag{15}$$

where $\mathcal{SN}_s(\cdot)$ is $\mathcal{SN}(\cdot)$ with the threshold $s \cdot u_{th}$. Note, $\text{SDSA}_3(\cdot)$ is inspired by the spiking self-attention $\mathcal{SN}(Q_SK_S^{\text{T}}V_S * s)$ in Zhou et al. (2023). Because $Q_SK_S^{\text{T}}V_S$ yield large integers, a scale factor $s$ for normalization is needed to avoid gradient vanishing in Zhou et al. (2023). In our SDSA-3, we directly merge the $s$ into the threshold of the spiking neuron to circumvent the multiplication by $s$. Further, in SDSA-4, we set the threshold as a learnable parameter.

Table 1: Performance on ImageNet-1K (Deng et al., 2009). The input crop is 224×224. *We obtain these results by employing distillation training on method in DeiT (Touvron et al., 2021). When trained directly, the accuracy are 78.0% ($T = 1$) and 79.7% ($T = 4$). Note, "Spike", "Para", and "Step" in all Table headers of this paper denote "Spike-driven", "Parameters", and "Timestep".

| Methods | Architecture | Spike | Param (M) | Power (mJ) | Step | Acc.(%) |
|---|---|---|---|---|---|---|
| ANN2SNN | ResNet-34 (Rathi et al., 2020) | ✓ | 21.8 | - | 250 | 61.5 |
| | VGG-16 (Wu et al., 2021) | ✓ | - | - | 16 | 65.1 |
| | VGG-16 (Hu et al., 2023) | ✓ | 138.4 | 44.9 | 7 | 73.0 |
| CNN-based SNN | SEW-Res-SNN | ✗ | 25.6 | 4.9 | 4 | 67.8 |
| | (Fang et al., 2021) | ✗ | 60.2 | 12.9 | 4 | 69.2 |
| | MS-Res-SNN | ✓ | 21.8 | 5.1 | 4 | 69.4 |
| | (Hu et al., 2024) | ✓ | 77.3 | 10.2 | 4 | 75.3 |
| | Att-MS-Res-SNN | ✗ | 22.1 | 0.6 | 1 | 69.2 |
| | (Yao et al., 2023d) | ✗ | 78.4 | 7.3 | 4 | 76.3 |
| ANN | RSB-CNN-152 (Wightman et al., 2021) | ✗ | 60 | 53.4 | 1 | 81.8 |
| | ViT (Dosovitskiy et al., 2021) | ✗ | 86 | 81.0 | 1 | 79.7 |
| Transformer -based SNN | SpikFormer | ✗ | 29.7 | 11.6 | 4 | 73.4 |
| | (Zhou et al., 2023) | ✗ | 66.3 | 21.5 | 4 | 74.8 |
| | Spike-driven Transformer | ✓ | 29.7 | 4.5 | 4 | 74.6 |
| | (Yao et al., 2023b) | ✓ | 66.3 | 6.1 | 4 | 76.3 |
| | Meta-SpikeFormer **(This Work)** | ✓ | 15.1 | 4.0 | 1 | 71.8 |
| | | ✓ | 15.1 | 16.7 | 4 | 74.1 |
| | | ✓ | 31.3 | 7.8 | 1 | 75.4 |
| | | ✓ | 31.3 | 32.8 | 4 | 77.2 |
| | | ✓ | 55.4 | 13.0 | 1 | 79.1* |
| | | ✓ | 55.4 | 52.4 | 4 | **80.0*** |

**Shortcuts.** Residual learning in SNNs mainly considers two points: first, whether identity mapping (He et al., 2016a) can be realized, which determines whether there is a degradation problem; second, whether spike-driven computing can be guaranteed, which is the basis of SNNs' low-power. There are three shortcuts in SNNs, see Fig. 4. Vanilla Shortcut (VS) (Zheng et al., 2021) execute a shortcut between membrane potential and spike that are consistent with those in Res-CNN (He et al., 2016b). It can be spike-driven, but cannot achieve identity mapping (Fang et al., 2021). Spike-Element-Wise (SEW) (Fang et al., 2021) exploits a shortcut to connect spikes in different layers. Identity mapping is possible with SEW, but spike addition results in integers.

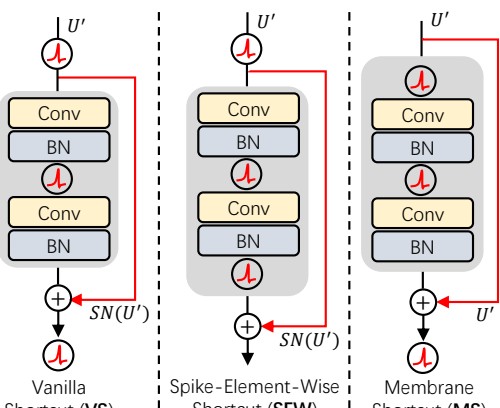

Figure 4: Existing shortcut in SNNs.

*Thus, SEW-SNN is an "integer-driven" rather than a spike-driven SNN.* Membrane Shortcut (MS) makes a shortcut between the membrane potential of neurons, and can achieve identity mapping with spike-driven (Hu et al., 2024). We use MS in this work and report the accuracy of other shortcuts.

## 4 EXPERIMENTS

In the classification task, we set the timestep $T = 1$ for 200 epoch training to reduce training cost, then finetune it to $T = 4$ with an additional 20 epochs. We here mainly emphasize the network scale. Other details, such as training schemes and configurations, are in Appendix C.1. Moreover, we use the model trained on ImageNet classification to finetune the detection or segmentation heads. *This is the first time that the SNN domain has been able to process dense prediction tasks in a **unified** way.*

Table 2: Performance of event-based action recognition on HAR-DVS (Wang et al., 2022).

| Methods | Architecture | Spike | Param(M) | Acc.(%) |
|---|---|---|---|---|
| ANN | SlowFast (Feichtenhofer et al., 2019) | ✗ | 33.6 | 46.5 |
| | ACTION-Net (Wang et al., 2021b) | ✗ | 27.9 | 46.9 |
| | TimeSformer (Bertasius et al., 2021) | ✗ | 121.2 | 50.8 |
| CNN-based SNN | Res-SNN-34 (Fang et al., 2021) | ✗ | 21.8 | 46.1 |
| Transformer-based SNN | Meta-SpikeFormer (**This Work**) | ✓ | 18.3 | **47.5** |

## 4.1 IMAGE CLASSIFICATION

**Setup.** ImageNet-1K (Deng et al., 2009) is one of computer vision's most widely used datasets. It contains about 1.3M training and 50K validation images, covering common 1K classes. As shown in Fig. 2, changing the channel number can obtain various model scales. We set $C = 32, 48, 64$. Their parameters are 15.1M, 31.3M, and 55.4M, respectively. The energy cost of ANNs is FLOPs times $E_{MAC}$. The energy cost of SNNs is FLOPs times $E_{AC}$ times spiking firing rate. $E_{MAC} = 4.6pJ$ and $E_{AC} = 0.9pJ$ are the energy of a MAC and an AC, respectively (more details in Appendix B).

**Results.** We comprehensively compare Meta-SpikeFormer with other methods in accuracy, parameter, and power (Table 1). We can see that Meta-SpikeFormer achieves SOTA in the SNN domain with significant accuracy advantages. For example, **Meta-SpikeFormer** vs. MS-Res-SNN vs. Spike-driven Transformer: Param, **55M** vs. 77M vs. 66M; Acc, **79.7%** vs. 75.3% vs. 76.3%. If we employ the distillation strategy in DeiT (Touvron et al., 2021), the accuracy of 55M Meta-SpikeFormer at $T = 1$ and $T = 4$ can be improved to 79.1% and 80.0%. It should be noted that after adding more Conv layers at stage 1/2, the power of Meta-SpikeFormer increases. This is still very cost-effective. For instance, **Meta-SpikeFormer** vs. MS-Res-SNN vs. Spike-driven Transformer: Power, **11.9mJ** ($T = 1$) vs. 6.1mJ ($T = 4$) vs. 10.2mJ ($T = 4$); Acc, **79.1%** vs. 75.3% vs. 76.3%. In summary, Meta-SpikeFormer obtained the first achievement of 80% accuracy on ImageNet-1K in SNNs.

## 4.2 EVENT-BASED ACTIVITY RECOGNITION

Event-based vision (also known as "neuromorphic vision") is one of the most typical application scenarios of neuromorphic computing (Indiveri & Douglas, 2000; Gallego et al., 2022; Wu et al., 2022). The famous neuromorphic camera, Dynamic Vision Sensors (DVS) (Lichtsteiner et al., 2008), encodes vision information into a sparse event (spike with address) stream only when brightness changes. Since the spike-driven nature, SNNs have the inherent advantages of low power and low latency when processing events. We use HAR-DVS to evaluate our method. HAR-DVS (Wang et al., 2022) is the largest event-based Human Activity Recognition (HAR) dataset currently, containing 300 classes and 107,646 samples, acquired by a DAVIS346 camera with a spatial resolution of $346 \times 260$. The raw HAR-DVS is more than 4TB, and the authors convert each sample into frames to form a new HAR-DVS. We handle HAR-DVS in a direct training manner with $T = 8$. Meta-SpikeFormer achieves comparable accuracy to ANNs and is better than the Conv-based SNN baseline (Table 2).

## 4.3 OBJECT DETECTION

So far, no backbone with direct training in SNNs can handle classification, detection, and segmentation tasks concurrently. Only recently did the SNN domain have the first directly trained model to process detection (Su et al., 2023). We evaluate Meta-SpikeFormer on the COCO benchmark (Lin et al., 2014) that includes 118K training images (train2017) and 5K validation images (val2017). We first transform *mmdetection* (Chen et al., 2019) codebase into a spiking version and use it to execute our model. We exploit Meta-SpikeFormer with Mask R-CNN (He et al., 2017). ImageNet pre-trained weights are utilized to initialize the backbones, and Xavier (Glorot & Bengio, 2010) to initialize the added layers. Results are shown in Table 3. We can see that Meta-SpikeFormer achieves SOTA results in the SNN domain. Note, EMS-Res-SNN got performance close to ours using 26.9M parameters, thanks to its direct training strategy and special network design. In contrast, we only use a fine-tuning strategy, which results in lower design and training costs. To be fair, we also tested directly trained Meta-SpikeFormer + Yolo and achieved good performance (Appendix C.2).

Table 3: Performance of object detection on COCO val2017 (Lin et al., 2014).

| Methods | Architecture | Spike | Param(M) | Power(mJ) | Step | mAP@0.5(%) |
|---|---|---|---|---|---|---|
| ANN | ResNet-18 (Yu et al., 2022a) | ✗ | 31.2 | 890.6 | 1 | 54.0 |
| | PVT-Tiny (Wang et al., 2021a) | ✗ | 32.9 | 1040.5 | 1 | 59.2 |
| ANN2SNN | Spiking-Yolo (Kim et al., 2020) | ✓ | 10.2 | - | 3500 | 25.7 |
| | Spike Calibration (Li et al., 2022) | ✓ | 17.1 | - | 512 | 45.3 |
| CNN-based SNN | Spiking Retina (Zhang et al., 2023) | ✗ | 11.3 | - | 4 | 28.5 |
| | EMS-Res-SNN (Su et al., 2023) | ✓ | 26.9 | - | 4 | 50.1 |
| Transformer -based SNN | Meta-SpikeFormer (**This Work**) | ✓ | 34.9 | 49.5 | 1 | 44.0 |
| | | ✓ | 75.0 | 140.8 | 1 | 51.2 |

Table 4: Performance of semantic segmentation on ADE20K (Zhou et al., 2017).

| Methods | Architecture | Spike | Param(M) | Power(mJ) | Step | MIoU(%) |
|---|---|---|---|---|---|---|
| ANN | ResNet-18 (Yu et al., 2022a) | ✗ | 15.5 | 147.1 | 1 | 32.9 |
| | PVT-Tiny (Wang et al., 2021a) | ✗ | 17.0 | 152.7 | 1 | 35.7 |
| | PVT-Small (Wang et al., 2021a) | ✗ | 28.2 | 204.7 | 1 | 39.8 |
| | DeepLab-V3 (Zhang et al., 2022a) | ✗ | 68.1 | 1240.6 | 1 | 42.7 |
| Transformer -based SNN | Meta-SpikeFormer (**This Work**) | ✓ | 16.5 | 22.1 | 1 | 32.3 |
| | | ✓ | 16.5 | 88.1 | 4 | 33.6 |
| | | ✓ | 58.9 | 46.6 | 1 | 34.8 |
| | | ✓ | 58.9 | 183.6 | 4 | 35.3 |

## 4.4 SEMANTIC SEGMENTATION

ADE20K (Zhou et al., 2017) is a challenging semantic segmentation benchmark commonly used in ANNs, including 20K and 2K images in the training and validation set, respectively, and covering 150 categories. *No SNN has yet reported processing results on ADE20K*. In this work, Meta-SpikeFormers are evaluated as backbones equipped with Semantic FPN (Kirillov et al., 2019). ImageNet trained checkpoints are used to initialize the backbones while Xavier (Glorot & Bengio, 2010) is utilized to initialize other newly added layers. We transform *mmsegmentation* (Contributors, 2020) codebase into a spiking version and use it to execute our model. Training details are given in Appendix C.3. We see that in lightweight models (16.5M in Table 4), Meta-SpikeFormer with lower power achieves comparable results to ANNs. For example, **Meta-SpikeFormer** ($T = 1$) vs. ResNet-18 vs. PVT-Tiny: Param, **16.5M** vs. 15.5M vs. 17.0M; MIoU, **32.3%** vs. 32.9% vs. 35.7%; Power, **22.1mJ** vs. 147.1mJ vs. 152.7mJ. To demonstrate the superiority of our method over other SNN segmentation methods, we also evaluate our method on VOC2012 and achieve SOTA results (Appendix C.4).

## 4.5 ABLATION STUDIES

**Conv-based SNN Block.** In this block, We follow the ConvFormer in (Yu et al., 2022b), which uses SpeConv as token mixer in stage-1/2. However, we note that SpeConv in Meta-SpikeFormer seems less important. After removing SpeConv, the power is reduced by 29.5%, the accuracy is only lost by 0.3%. If we replace the channel Conv with the channel MLP in (Yu et al., 2022b), the accuracy will drop by up to 2%. Thus, the design of Conv-based SNN Blocks is important to SNNs' performance. Moreover, we experimentally verified (specific results are omitted) that keeping only one stage of the Conv-based block or using only four Conv layers leads to lower performance on downstream tasks .

**Transformer-based SNN Block.** Spike-driven Transformer in (Yao et al., 2023b) uses linear layers (i.e., $1 \times 1$ convolution) to generate $Q_S, K_S, V_S$. We find that replacing linear with RepConv can improve accuracy and reduce the parameter number, but energy costs will increase. The design of the SDSA operator and pyramid structure will also affect task accuracy. Overall, SDSA-3 has the highest computational complexity (more details in Appendix A), and its accuracy is also the best.

**Shortcut.** In our architecture, MS has the highest accuracy. Shortcut has almost no impact on power.

Table 5: Ablation studies of Meta-SpikeFormer on ImageNet-1K. In each ablation experiment, we start with Meta-SpikeFormer ($C = 48$) with $T = 1$ as the baseline and modify just one point to track how the parameters, power, accuracy vary. * Does not converge.

| Ablation | Methods | Param(M) | Power(mJ) | Acc.(%) |
|---|---|---|---|---|
| | **Meta-SpikeFormer (Baseline)** | **31.3** | **7.8** | **75.4** |
| Conv-based SNN Block | Remove SepConv | 30.9 | 5.5 | 75.1 (-0.3) |
| | Channel Conv –> Channel MLP | 25.9 | 6.3 | 73.4 (-2.0) |
| | Stage 1 –> $2C \times \frac{H}{4} \times \frac{W}{4}$ | 31.8 | 7.4 | 75.2 (-0.2) |
| Transformer-based SNN block | RepConv-1/2/3 –> Linear | 27.2 | 6.0 | 75.0 (-0.4) |
| | RepConv-4 –> Linear | 30.0 | 7.1 | 75.3 (-0.1) |
| | SDSA-3 –> SDSA-1 | 31.3 | 7.2 | 74.6 (-0.8) |
| | SDSA-3 –> SDSA-2 | 28.6 | 6.3 | 74.2 (-1.2) |
| | SDSA-3 –> SDSA-4 | 31.3 | 7.7 | 75.4 (+0.0) |
| Shortcut | Membrane Shortcut –> Vanilla shortcut | 31.3 | - | * |
| | Membrane Shortcut –> SEW shortcut | 31.3 | 7.8 | 73.5 (-1.9) |
| Architecture | Remove Pyramid (Stage4 = Stage 3) | 26.9 | 7.4 | 74.7 (-0.7) |
| | Fully CNN-based SNN blocks | 36.0 | 2.9 | 72.5 (-2.9) |
| | Fully Transformer-based SNN blocks | 26.2 | 5.4 | 71.7 (-3.7) |

**Architecture.** We change the network to fully Conv-based or Transformer-based blocks. Performance is significantly reduced in both cases. We note that compared to Meta-SpikeFormer, the power of fully spiking Transformer and spiking CNN are reduced. These observations can inspire future architectural designs to achieve multiple trade-offs in terms of parameter, power, and accuracy.

## 5 DISCUSSION AND CONCLUSION

**Discussion: How does Meta-SpikeFormer inspire future neuromorphic chip design?** The *technical* inspiration of Meta-SpikeFormer for chip design lies in *three* folds: i) *Conv+ViT design*. This hybrid progressive local-global modeling can leverage the strengths of both CNNs and Transformers (Guo et al., 2022), where the former models features and the latter captures long-range dependencies. We experimentally verify that this design is beneficial to the performance and versatility of SNNs. ii) *SDSA operator* is the core design of long-distance dependency modeling in Transformer-based SNN block, but this is a design that current neuromorphic chips lack. iii) *Meta architecture*. Given meta Conv-based and Transformer-based blocks, researchers can perform targeted optimization of the design details inside the meta SNN blocks according to their requirements in terms of accuracy, parameters, and power. As shown by our ablation experiments in Table 5.

The *significance* of Meta-SpikeFormer to chip design lies in *three* folds: i) *Algorithm-hardware co-design*. Most neuromorphic chip design begins from the bottom of the compute stack, i.e., the materials and devices (Schuman et al., 2022). The excellent features shown by our algorithm may attract and inspire algorithm-driven chip design. ii) *Confidence in large-scale neuromorphic computing*. Small-scale neuromorphic computing has shown significant power and performance advantages (Yin et al., 2021; Rao et al., 2022). We demonstrate the potential of larger-scale SNNs in performance and versatility. iii) *Reduce chip design costs*. Meta design facilitates follow-up optimization by subsequent researchers, helps the SNN field to quickly narrow the gap with ANNs, and reduces the cost of algorithm exploration required before algorithm-driven hardware design.

**Conclusion.** This paper investigates the meta design of Transformer-based SNNs, involving architecture, spike-driven self-attention, shortcut, etc. The proposed Meta-SpikeFormer is the first direct training SNN backbone that can perform classification, detection, and segmentation tasks concurrently, and we achieve state-of-the-art results on all tested datasets. Remarkably, for the first time, we advanced the accuracy of the SNN domain on ImageNet-1K to 80%, which is 3.7% higher than the prior SOTA result with 17% fewer parameters. This work paves the way for SNN to serve as a universal vision backbone and can inspire future Transformer-based neuromorphic chip designs.

ACKNOWLEDGEMENT

This work was partially supported by National Science Foundation for Distinguished Young Scholars (62325603), National Natural Science Foundation of China (62236009, U22A20103), Beijing Natural Science Foundation for Distinguished Young Scholars (JQ21015), and CAAI-MindSpore Open Fund, developed on OpenI Community.

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

**Limitations** of this work are larger scale models, more vision tasks, further optimization of accuracy, power and parameter amount, optimization of training consumption caused by multiple timesteps, etc., and we will work on them in future work. The experimental results in this paper are reproducible. We explain the details of model training and configuration in the main text and supplement it in the appendix. Our codes and models of Meta-SpikeFormer will be available on GitHub after review. Moreover, in this work, the designed meta SNN architecture is tested on vision tasks. For language tasks, the challenges faced will be different, such as parallel spiking neuron design, long-term dependency modeling in the temporal dimension, pre-training, architecture design, etc. need to be considered. This work can at least provide positive inspiration for SNN processing language tasks in long-term dependency modeling and architecture design, and we are working in this direction.

## A  SPIKE-DRIVEN SELF-ATTENTION (SDSA) OPERATORS

In this Section, we understand vanilla and spike-driven self-attention from the perspective of computational complexity.

### A.1  VANILLA SELF-ATTENTION (VSA)

Given a float-point input sequence $X \in \mathbb{R}^{N \times D}$, float-point Query ($Q$), Key ($K$), and Value ($V$) in $\mathbb{R}^{N \times D}$ are calculated by three learnable linear matrices, where $N$ is the token number, $D$ is the channel dimension. The vanilla scaled dot-product self-attention is computed as (Dosovitskiy et al., 2021):

$$\text{VSA}(Q, K, V) = \text{softmax}\left(\frac{QK^{\text{T}}}{\sqrt{d}}\right) V, \tag{16}$$

where $d = D/H$ is the feature dimension of one head and $H$ is the head number, $\sqrt{d}$ is the scale factor. Generally, VSA performs multi-head self-attention, i.e., divide $Q, K, V$ into $H$ heads in the channel dimension. In the $i$-th head, $Q^i, K^i, V^i$ in $\mathbb{R}^{N \times D/H}$. After the self-attention operation is performed on the $H$ heads respectively, the outputs are concatenated together.

In VSA, $Q$ and $K$ are matrix multiplied first, and then their output is matrix multiplied with $V$. The computational complexity of $\text{VSA}(\cdot)$ is $O(N^2 D)$, which has a *quadratic* relationship with the toke number $N$.

### A.2  SPIKE-DRIVEN SELF-ATTENTION (SDSA)

In our Transformer-based SNN blocks, as shown in Fig. 2, given a spike input sequence $S \in \mathbb{R}^{T \times N \times D}$, spike-form (binary) $Q_S$, $K_S$, and $V_S$ in $\mathbb{R}^{T \times N \times D}$ are calculated by three learnable re-parameterization convolutions Ding et al. (2021) with $3 \times 3$ kernel size:

$$Q_S = \mathcal{SN}(\text{RepConv}_1(U)), K_S = \mathcal{SN}(\text{RepConv}_2(U)), V_S = \mathcal{SN}(\text{RepConv}_3(U)), \tag{17}$$

where $\text{RepConv}(\cdot)$ denotes the re-parameterization convolution, $\mathcal{SN}(\cdot)$ is the spiking neuron layer. For the convenience of mathematical expression, we assume $T = 1$ in the subsequent formulas.

**SDSA-1.** The leftmost SDSA-1 in Fig. 3 is the operator proposed in Spike-driven Transformer (Yao et al., 2023b). The highlight of SDSA-1 is that the matrix multiplication between $Q_S, K_S, V_S$ in SDSA is replaced by Hadamard product:

$$\text{SDSA}_1(Q_S, K_S, V_S) = Q_S \otimes \mathcal{SN}\left(\text{SUM}_\text{c}\left(K_S \otimes V_S\right)\right), \tag{18}$$

where $\otimes$ is the Hadamard product, $\text{SUM}_\text{c}(\cdot)$ represents the sum of each column, and its output is a $D$-dimensional row vector. The Hadamard product between spike tensors is equivalent to the mask operation. Compared to the VSA in Eq. 16, $\text{SUM}_\text{c}(\cdot)$ and $\mathcal{SN}(\cdot)$ take the role of softmax and scale.

Now, we analyze the computational complexity of SDSA-1. Before that, we would like to introduce the concept of *linear attention*. If the softmax in VSA is removed, $K$ and $V$ can be multiplied first, and then their output is matrix multiplied with $Q$. The computational complexity becomes $O(ND^2/H)$, which has a linear relationship with the toke number $N$. This variant of attention is called linear attention (Katharopoulos et al., 2020). Further, consider an extreme case in linear attention, set

$H = D$. That is, in each head, $Q^i, K^i, V^i$ in $\mathbb{R}^{N \times 1}$. Then, the computational complexity is $O(ND)$, which has a linear relationship with both the toke number $N$ and the channel dimension $D$. This variant of linear attention is called *hydra attention* (Bolya et al., 2023).

SDSA-1 has the same computational complexity as hydra attention, i.e., $O(ND)$. Firstly, $K_S$ and $V_S$ in Eq. 18 participate in the operation first, thus it is a kind of linear attention. Further, we consider the special operation of Hadamard product. Assume that the $i$-th column vectors in $K_S$ and $V_S$ are $a$ and $b$ respectively. Taking the Hadamard product of $a$ and $b$ and summing them is equivalent to multiplying $b$ times the transpose of $a$, i.e., $\mathrm{SUM_c}(a \otimes b) = a^T b$. In total, there are $D$ times of dot multiplication between vectors, and $N$ additions are performed each time. Thus, the computational complexity of SDSA-1 is $O(ND)$, which is consistent with hydra attention (Bolya et al., 2023).

**SDSA-2.** SDSA-1 in Eq. 18 actually first uses $Q_S$ and $K_S$ to calculate the binary self-attention scores, and then performs feature masking on $V_S$ in the channel dimension. We can also get the binary attention scores using only $Q_S$, i.e., SDSA-2 is presented as:

$$\mathrm{SDSA}_2(Q_S, V_S) = \mathcal{SN}\left(\mathrm{SUM_c}\left(Q_S\right)\right) \otimes V_S. \tag{19}$$

We evaluate SDSA-1 and SDSA-2 in Table 5. Specifically, SDSA-1-based Meta-SpikeFormer vs. SDSA-2-based Meta-SpikeFormer: Param, 31.3M vs. 28.6M; Power, 7.2mJ vs. 6.3mJ; Acc, 74.6% vs. 74.2%. It can be seen that with the support of the Meta-SpikeFormer architecture, even if the Key matrix $K_S$ is removed, the accuracy is only lost by 0.4%. The number of parameters and energy consumption are reduced by 8.7% and 12.5% respectively. Since the Hadamard product between spiking tensors $Q_S$ and $K_S$ in SDSA-1 can be regarded as a mask operation without energy cost, SDSA-1 and SDSA-2 have the same computational complexity, i.e., $O(ND)$. SDSA-2-based Meta-SpikeFormer has fewer parameters and power because there is no need to generate $K_S$.

**SDSA-3** is the spike-driven self-attention operator used by default in this work, which is presented as:

$$\mathrm{SDSA}_3(Q_S, K_S, V_S) = \mathcal{SN}_s\left(Q_S\left(K_S^\mathrm{T} V_S\right)\right) = \mathcal{SN}_s((Q_S K_S^\mathrm{T})V_S). \tag{20}$$

In theory, the time complexity of $Q_S(K_S^\mathrm{T} V_S)$ and $(Q_S K_S^\mathrm{T})V_S$ are $O(N^2 D)$ and $O(ND^2)$, respectively. The latter has a linear relationship with $N$, thus SDSA-3 is also a linear attention. Since $Q_S K_S^\mathrm{T} V_S$ yields large integers, a scale multiplication $s$ for normalization is needed to avoid gradient vanishing. In our SDSA-3, we incorporate the $s$ into the threshold of the spiking neuron to circumvent the multiplication by $s$. That is, the threshold in Eq. 20 is $s \cdot u_{th}$. We write such a spiking neuron layer with threshold $s \cdot u_{th}$ as $\mathcal{SN}_s(\cdot)$.

**SDSA-4.** On the basis of SDSA-3, we directly set the threshold of $\mathcal{SN}(\cdot)$ in Eq. 15 as a learnable parameter, and its initialization value is $s \cdot u_{th}$. We have experimentally found that the performance of SDSA-3 and SDSA-4 is almost the same (see Table 5). SDSA-4 consumes 0.1mJ less energy than SDSA-3 because the network spiking firing rate in SDSA-4 is slightly smaller than that in SDSA-3.

### A.3 DISCUSSION ABOUT SDSA OPERATORS

Compared with vanilla self-attention, the $Q_S, K_S, V_S$ matrices of spike-driven self-attention are in the form of binary spikes, and the operations between $Q_S, K_S, V_S$ do not include softmax and scale. Since there is no softmax and $K_S$ and $V_S$ can be computed first, spike-driven self-attention must be linear attention. This is the natural advantage of a spiking Transformer. On the other hand, in the current SDSA design, the operation between $Q_S, K_S, V_S$ is Hadamard product or matrix multiplication, both of which can be converted into sparse addition operations. Therefore, SDSA not only has low computational complexity, but also only has sparse addition. Its energy consumption is much lower than that of vanilla self-attention (see Appendix B).

In Yu et al. (2022a;b), the authors summarized various ViT variants and argued that there is general architecture abstracted from ViTs by not specifying the token mixer (self-attention). This paper experimentally verifies that this view also holds true in Transformer-based SNNs. In Table 5, we tested four SDSA operators and found that the performance changes between SDSA-1/2/3/4 were not large (less than 1.2%). We expect the SNN domain to design more powerful SDSA operators in the future, e.g., borrowing from Swin (Liu et al., 2021), hierarchical attention (Hatamizadeh et al., 2023), and so on.

Table 6: FLOPs of self-attention modules. The FLOPs in VSA and SDSA are multiplied by $E_{MAC} = 4.6pJ$ and $E_{AC} = 0.9pJ$ respectively to obtain the final energy cost. $R_C$, $\widehat{R}$ denote the sum of spike firing rates of various spike matrices.

| | VSA | SDSA-1 | SDSA-2 | SDSA-3 | SDSA-4 |
|---|---|---|---|---|---|
| $Q, K, V$ | $3ND^2$ | $T \cdot R_C \cdot 3 \cdot FL_{Conv}$ | $T \cdot R_C \cdot 2 \cdot FL_{Conv}$ | $T \cdot R_C \cdot 3 \cdot FL_{Conv}$ | $T \cdot R_C \cdot 3 \cdot FL_{Conv}$ |
| $f(Q, K, V)$ | $2N^2D$ | $T \cdot \widehat{R} \cdot ND$ | $T \cdot \widehat{R} \cdot ND$ | $T \cdot \widehat{R} \cdot ND^2$ | $T \cdot \widehat{R} \cdot ND^2$ |
| Scale | $N^2$ | - | - | - | - |
| Softmax | $2N^2$ | - | - | - | - |
| Linear | $FL_{MLP}$ | $T \cdot R_C \cdot FL_{Conv}$ | $T \cdot R_C \cdot FL_{Conv}$ | $T \cdot R_C \cdot FL_{Conv}$ | $T \cdot R_C \cdot FL_{Conv}$ |

## B  THEORETICAL ENERGY EVALUATION

### B.1  SPIKE-DRIVEN OPERATORS IN SNNS

Spike-driven operators for SNNs are fundamental to low-power neuromorphic computing. In CNN-based SNNs, spike-driven Conv and MLP constitute the entire network. Specifically, the matrix multiplication between the weight and spike matrix in spike-driven Conv and MLP is transformed into sparse addition, which is implemented as addressable addition in neuromorphic chips (Frenkel et al., 2023).

By contrast, $Q_S$, $K_S$, $V_S$ in spike-driven self-attention involve two matrix multiplications. One way is to execute element-wise multiplication between $Q_S$, $K_S$, $V_S$, like SDSA-1 in (Yao et al., 2023b) and SDSA-2 in this work (Eq. 19). And, element multiplication in SNNs is equivalent to mask operation with no energy cost. Another method is to perform multiplication directly between $Q_S$, $K_S$, $V_S$, which is then converted to sparse addition, like spike-driven Conv and MLP (SDSA-3/4 in this work).

### B.2  ENERGY CONSUMPTION OF META-SPIKEFORMER

When evaluating algorithms, the SNN field often ignores specific hardware implementation details and estimates theoretical energy consumption for a model (Panda et al., 2020; Yin et al., 2021; Yang et al., 2022; Yao et al., 2023d; Wang et al., 2023a). This theoretical estimation is just to facilitate the qualitative energy analysis of various SNN and ANN algorithms.

Theoretical energy consumption estimation can be performed in a simple way. For example, the energy cost of ANNs is FLOPs times $E_{MAC}$, and the energy cost of SNNs is FLOPs times $E_{AC}$ times network spiking firing rate. $E_{MAC} = 4.6pJ$ and $E_{AC} = 0.9pJ$ are the energy of a MAC and an AC, respectively, in 45nm technology (Horowitz, 2014).

There is also a more refined method of evaluating energy consumption for SNNs. We can count the spiking firing rate of each layer, and then the energy consumption of each layer is FLOPs times $E_{AC}$ times the layer spiking firing rate. The nuance is that the network structure affects the number of additions triggered by a single spike. For example, the energy consumption of the same spike tensor differs when doing matrix multiplication with various convolution kernel sizes.

In this paper, we count the spiking firing rate of each layer, then estimate the energy cost. Specifically, the FLOPs of the $n$-th Conv layer in ANNs Molchanov et al. (2017) are:

$$FL_{Conv} = (k_n)^2 \cdot h_n \cdot w_n \cdot c_{n-1} \cdot c_n, \tag{21}$$

where $k_n$ is the kernel size, $(h_n, w_n)$ is the output feature map size, $c_{n-1}$ and $c_n$ are the input and output channel numbers, respectively. The FLOPs of the $m$-th MLP layer in ANNs are:

$$FL_{MLP} = i_m \cdot o_m, \tag{22}$$

where $i_m$ and $o_m$ are the input and output dimensions of the MLP layer, respectively.

For spike-driven Conv or MLP, we only need to consider additional timestep $T$ and layer spiking firing rates. The power of spike-driven Conv and MLP are $E_{AC} \cdot T \cdot R_C \cdot FL_{Conv}$ and $E_{AC} \cdot T \cdot R_M \cdot FL_{MLP}$ respectively. $R_C$ and $R_M$ represent the layer spiking firing rate, defined as the proportion of non-zero

elements in the spike tensor. For the SDSA modules in Fig. 3, the energy cost of the Rep-Conv part is consistent with spike-driven Conv. The energy cost of the SDSA operator part is given in Table 6. Combining Table 5, we observe that the $SDSA(\cdot)$ function itself does not consume much energy because the $Q$, $K$, and $V$ matrices themselves are sparse. The evidence is that SDSA-1 saves about 0.6mJ of energy consumption compared to SDSA-3 (see Table 5). In order to give readers an intuitive feeling about the spiking firing rate, we give the detailed spiking firing rates of a Meta-SpikeFormer model in Table 11.

# C  DETAILED CONFIGURATIONS AND HYPER-PARAMETER OF META-SPIKEFORMER MODELS

## C.1  IMAGENET-1K EXPERIMENTS

On ImageNet-1K classification benchmark, we employ three scales of Meta-SpikeFormer in Table 7 and utilize the hyper-parameters in Table 8 to train models in our paper.

Table 7: Configurations of different Meta-SpikeFormer models.

| stage | # Tokens | Layer Specification | | | 15M | 31M | 55M |
|---|---|---|---|---|---|---|---|
| 1 | $\frac{H}{2} \times \frac{W}{2}$ | Downsampling | | Conv | 7x7 stride 2 | | |
| | | | | Dim | 32 | 48 | 64 |
| | | Conv-based SNN block | SepConv | DWConv | 7x7 stride 1 | | |
| | | | | MLP ratio | 2 | | |
| | | | Channel Conv | Conv | 3x3 stride 1 | | |
| | | | | Conv ratio | 4 | | |
| | $\frac{H}{4} \times \frac{W}{4}$ | Downsampling | | Conv | 3x3 stride 2 | | |
| | | | | Dim | 64 | 96 | 128 |
| | | Conv-based SNN block | SepConv | DWConv | 7x7 stride 1 | | |
| | | | | MLP ratio | 2 | | |
| | | | Channel Conv | Conv | 3x3 stride 1 | | |
| | | | | Conv ratio | 4 | | |
| 2 | $\frac{H}{8} \times \frac{W}{8}$ | Downsampling | | Conv | 3x3 stride 2 | | |
| | | | | Dim | 128 | 192 | 256 |
| | | Conv-based SNN block | SepConv | DWConv | 7x7 stride 1 | | |
| | | | | MLP ratio | 2 | | |
| | | | Channel Conv | Conv | 3x3 stride 1 | | |
| | | | | Conv ratio | 4 | | |
| | | | # Blocks | | 2 | | |
| 3 | $\frac{H}{16} \times \frac{W}{16}$ | Downsampling | | Conv | 3x3 stride 2 | | |
| | | | | Dim | 256 | 384 | 512 |
| | | Transformer-based SNN block | SDSA | RepConv | 3x3 stride 1 | | |
| | | | Channel MLP | MLP ratio | 4 | | |
| | | | # Blocks | | 6 | | |
| 4 | $\frac{H}{16} \times \frac{W}{16}$ | Downsampling | | Conv | 3x3 stride 1 | | |
| | | | | Dim | 360 | 480 | 640 |
| | | Transformer-based SNN block | SDSA | RepConv | 3x3 stride 1 | | |
| | | | Channel MLP | MLP ratio | 4 | | |
| | | | # Blocks | | 2 | | |

## C.2  COCO EXPERIMENTS

In this paper, we have used two methods to utilize Meta-SpikeFormer for object detection. We first exploit Meta-SpikeFormer as backbones for object detection, fine-tuning for 24 epochs after inserting the Mask R-CNN detector (He et al., 2017). The batch size is 12. The AdamW is employed with an initial learning rate of $1 \times 10^{-4}$ that will decay in the polynomial decay schedule with a power of 0.9. Images are resized and cropped into $1333 \times 800$ for training and testing and maintain the ratio. Random horizontal flipping and resize with a ratio of 0.5 was applied for augmentation during

Table 8: Hyper-parameters for image classification on ImageNet-1K

| Hyper-parameter | Directly Training | Finetune |
|---|---|---|
| Model size | 15M/31M/55M | 15M/31M/55M |
| Timestemp | 1 | 4 |
| Epochs | 200 | 20 |
| Resolution | 224*224 | |
| Batch size | 1568 | 336 |
| Optimizer | LAMB | |
| Base Learning rate | 6e-4 | 2e-5 |
| Learning rate decay | Cosine | |
| Warmup eopchs | 10 | 2 |
| Weight decay | 0.05 | |
| Rand Augment | 9/0.5 | |
| Mixup | None | |
| Cutmix | None | |
| Label smoothing | 0.1 | |

training. This pre-training fine-tuning method is a commonly used strategy in ANNs. We use this method and get SOTA results (see Table 3), but with many parameters. To address this problem, we then train Meta-SpikeFormer in a direct training manner in conjunction with the lightweight Yolov5 [1] detector, which Yolov5 is re-implemented by us in a spike-driven manner. Results are reported in Table 9. The current SOTA result in SNNs on COCO is EMS-Res-SNN (Su et al., 2023), which improves the structure. We get better performance using parameters that are close to EMS-Res-SNN.

Table 9: Performance of object detection on COCO val2017 (Lin et al., 2014)

| Methods | Architecture | Spike-driven | Param (M) | Power (mJ) | Time Step | mAP@0.5 (%) |
|---|---|---|---|---|---|---|
| Conv-based SNN | EMS-Res-SNN (Su et al., 2023) | ✓ | 26.9 | - | 4 | 50.1 |
| Transformer-based SNN | Meta-SpikeFormer + Yolo | ✓ | 16.8 | 34.8 | 1 | 45.0 |
| | (**This Work**) | ✓ | 16.8 | 70.7 | 4 | 50.3 |

## C.3 ADE20K EXPERIMENTS

Meta-SpikeFormer is employed as the backbone equipped with Sementic FPN Lin et al. (2017), which is re-implemented in a spike-driven manner. In $T = 1$, ImageNet-1K trained checkpoints are used to initialize the backbones while Xavier is utilized to initialize other newly added SNN layers. We train the model for 160K iterations with a batch size of 20. The AdamW is employed with an initial learning rate of $1 \times 10^{-3}$ that will decay in the polynomial decay schedule with a power of 0.9. Then we finetuned the model to $T = 4$ and decreased the learning rate to $1 \times 10^{-4}$. To speed up training, we warm up the model for 1.5k iterations with a linear decay schedule.

## C.4 ADDITIONAL RESULTS ON VOC2012 SEGMENTATION

VOC2012 (Everingham et al., 2010) is a benchmark for segmentation which has 1460 and 1456 images in the training and validation set respectively, and covering 21 categories. Previous work using SNN for segmentation has used this dataset. Thus we also test our method on this dataset. We train the Meta-SpikeFormer for 80k iterations in $T = 1$ with ImageNet-1k trained checkpoints to initialize the backbones while Xavier is utilized to initialize other newly added SNN layers. Then we finetune the model to $T = 4$ with lower learning rate $1 \times 10^{-4}$. Other experiment settings are the same as the ADE20k benchmark. Results are given in Table 10, and we achieve SOTA results.

---

[1]https://github.com/ultralytics/yolov5

Table 10: Performance of semantic segmentation on VOC2012 (Everingham et al., 2010)

| Methods | Architecture | Spike -driven | Param (M) | Power (mJ) | Time Step | MIoU(%) |
|---------|--------------|---------------|-----------|------------|-----------|---------|
| ANN | FCN-R50 (Long et al., 2015) | ✗ | 49.5 | 909.6 | 1 | 62.2 |
| | DeepLab-V3 (Chen et al., 2017) | ✗ | 68.1 | 1240.6 | 1 | 66.7 |
| ANN2SNN | Spike Calibration (Li et al., 2022) | ✓ | - | - | 64 | 55.0 |
| CNN-based SNN | Spiking FCN (Kim et al., 2022) | ✓ | 49.5 | 383.5 | 20 | 9.9 |
| | Spiking DeepLab (Kim et al., 2022) | ✓ | 68.1 | 523.2 | 20 | 22.3 |
| Transformer -based SNN | Meta-SpikeFormer **(This Work)** | ✓ ✓ | 16.5 58.9 | 81.4 179.8 | 4 4 | 58.1 **61.1** |

Table 11: Layer spiking firing rates of model Meta-SpikeFormer ($T = 4$, 31.3M, SDSA-3) on ImageNet-1K.

| | | | $T=1$ | $T=2$ | $T=3$ | $T=4$ | Average |
|---|---|---|---|---|---|---|---|
| **Stage 1** | Downsampling | Conv | 1 | 1 | 1 | 1 | 1 |
| | ConvBlock SepConv | PWConv1 | 0.2662 | 0.4505 | 0.3231 | 0.4742 | 0.3785 |
| | | DWConv&PWConv2 | 0.3517 | 0.4134 | 0.3906 | 0.4057 | 0.3903 |
| | Channel Conv | Conv1 | 0.3660 | 0.5830 | 0.4392 | 0.5529 | 0.4852 |
| | | Conv2 | 0.1601 | 0.1493 | 0.1662 | 0.1454 | 0.1552 |
| | Downsampling | Conv | 0.4408 | 0.4898 | 0.4929 | 0.4808 | 0.4761 |
| | ConvBlock SepConv | PWConv1 | 0.2237 | 0.3658 | 0.3272 | 0.3544 | 0.3178 |
| | | DWConv&PWConv2 | 0.2276 | 0.2672 | 0.2590 | 0.2567 | 0.2526 |
| | Channel Conv | Conv1 | 0.3324 | 0.4640 | 0.4275 | 0.4433 | 0.4168 |
| | | Conv2 | 0.0866 | 0.0838 | 0.0811 | 0.0775 | 0.0823 |
| **Stage 2** | Downsampling | Conv | 0.3456 | 0.3916 | 0.3997 | 0.3916 | 0.3821 |
| | ConvBlock SepConv | PWConv1 | 0.2031 | 0.3845 | 0.3306 | 0.3648 | 0.3207 |
| | | DWConv&PWConv2 | 0.1860 | 0.2101 | 0.2020 | 0.1988 | 0.1992 |
| | Channel Conv | Conv1 | 0.2871 | 0.4499 | 0.4013 | 0.4233 | 0.3904 |
| | | Conv2 | 0.0548 | 0.0541 | 0.0501 | 0.0464 | 0.0513 |
| | ConvBlock SepConv | PWConv1 | 0.3226 | 0.4245 | 0.4132 | 0.4158 | 0.3940 |
| | | DWConv&PWConv2 | 0.1051 | 0.1051 | 0.1025 | 0.0995 | 0.1030 |
| | Channel Conv | Conv1 | 0.2863 | 0.3787 | 0.3732 | 0.3728 | 0.3528 |
| | | Conv2 | 0.0453 | 0.0418 | 0.0408 | 0.0382 | 0.0415 |
| **stage3** | Downsampling | Conv | 0.3817 | 0.4379 | 0.4436 | 0.4401 | 0.4259 |
| | Block1 SDSA | RepConv-1/2/3 | 0.1193 | 0.2926 | 0.2396 | 0.2722 | 0.2309 |
| | | $Q_S$ | 0.2165 | 0.2402 | 0.2377 | 0.2213 | 0.2289 |
| | | $K_S$ | 0.0853 | 0.0931 | 0.0935 | 0.0818 | 0.0884 |
| | | $V_S$ | 0.0853 | 0.1414 | 0.1227 | 0.1234 | 0.1182 |
| | | $K_S^T V_S$ | 0.3083 | 0.4538 | 0.4238 | 0.4023 | 0.3971 |
| | | $Q_S(K_S^T V_S)$ | 0.7571 | 0.8832 | 0.8674 | 0.8426 | 0.8376 |
| | | RepConv-4 | 0.4115 | 0.6402 | 0.6034 | 0.5398 | 0.5487 |
| | Channel MLP | Linear 1 | 0.2147 | 0.3849 | 0.3263 | 0.3637 | 0.3224 |
| | | Linear 2 | 0.0353 | 0.0298 | 0.0262 | 0.0232 | 0.0286 |
| | Block2 SDSA | RepConv-1/2/3 | 0.2643 | 0.4093 | 0.3706 | 0.3918 | 0.3590 |
| | | $Q_S$ | 0.1594 | 0.1859 | 0.1913 | 0.1871 | 0.1809 |
| | | $K_S$ | 0.0774 | 0.1029 | 0.1061 | 0.1034 | 0.0975 |
| | | $V_S$ | 0.0852 | 0.1271 | 0.1228 | 0.1232 | 0.1146 |
| | | $K_S^T V_S$ | 0.4125 | 0.5852 | 0.5805 | 0.5835 | 0.5404 |
| | | $Q_S(K_S^T V_S)$ | 0.8246 | 0.9216 | 0.9231 | 0.9190 | 0.8970 |
| | | RepConv-4 | 0.4148 | 0.6622 | 0.6737 | 0.6545 | 0.6013 |
| | Channel MLP | Linear 1 | 0.2899 | 0.4026 | 0.3756 | 0.3884 | 0.3641 |
| | | Linear 2 | 0.0302 | 0.0269 | 0.0239 | 0.0219 | 0.0258 |
| | Block3 SDSA | RepConv-1/2/3 | 0.2894 | 0.3877 | 0.3706 | 0.3773 | 0.3562 |
| | | $Q_S$ | 0.1419 | 0.1397 | 0.1437 | 0.1405 | 0.1415 |
| | | $K_S$ | 0.0590 | 0.0609 | 0.0639 | 0.0616 | 0.0614 |
| | | $V_S$ | 0.0904 | 0.1232 | 0.1279 | 0.1261 | 0.1169 |
| | | $K_S^T V_S$ | 0.3674 | 0.4703 | 0.4825 | 0.4863 | 0.4516 |
| | | $Q_S(K_S^T V_S)$ | 0.8423 | 0.8912 | 0.9010 | 0.8961 | 0.8827 |
| | | RepConv-4 | 0.3613 | 0.4850 | 0.5281 | 0.5072 | 0.4704 |
| | Channel MLP | Linear 1 | 0.3047 | 0.3795 | 0.3676 | 0.3727 | 0.3561 |
| | | Linear 2 | 0.0274 | 0.0248 | 0.0227 | 0.0211 | 0.0240 |

Continued on next page

**Table 11 – continued from previous page**

|  |  |  | $T=1$ | $T=2$ | $T=3$ | $T=4$ | Average |
|---|---|---|---|---|---|---|---|
|  |  | RepConv-1/2/3 | 0.2833 | 0.3469 | 0.3400 | 0.3430 | 0.3283 |
|  |  | $Q_S$ | 0.1910 | 0.1884 | 0.1937 | 0.1893 | 0.1906 |
|  |  | $K_S$ | 0.0570 | 0.0620 | 0.0658 | 0.0642 | 0.0622 |
|  | SDSA | $V_S$ | 0.0834 | 0.0986 | 0.1065 | 0.1043 | 0.0982 |
| Block4 |  | $K_S^T V_S$ | 0.3421 | 0.4375 | 0.4566 | 0.4670 | 0.4258 |
|  |  | $Q_S(K_S^T V_S)$ | 0.8279 | 0.8925 | 0.9067 | 0.9097 | 0.8842 |
|  |  | RepConv-4 | 0.3632 | 0.4932 | 0.5457 | 0.5365 | 0.4847 |
|  | Channel MLP | Linear 1 | 0.3040 | 0.3562 | 0.3487 | 0.3512 | 0.3400 |
|  |  | Linear 2 | 0.0282 | 0.0267 | 0.0244 | 0.0230 | 0.0256 |
|  |  | RepConv-1/2/3 | 0.2882 | 0.3334 | 0.3280 | 0.3298 | 0.3198 |
|  |  | $Q_S$ | 0.1577 | 0.1487 | 0.1501 | 0.1482 | 0.1512 |
|  |  | $K_S$ | 0.0440 | 0.0496 | 0.0528 | 0.0534 | 0.0499 |
|  | SDSA | $V_S$ | 0.0853 | 0.1276 | 0.1363 | 0.1377 | 0.1217 |
| Block5 |  | $K_S^T V_S$ | 0.3633 | 0.4934 | 0.5187 | 0.5365 | 0.4780 |
|  |  | $Q_S(K_S^T V_S)$ | 0.8424 | 0.9031 | 0.9178 | 0.9213 | 0.8961 |
|  |  | RepConv-4 | 0.3550 | 0.5158 | 0.5620 | 0.5678 | 0.5001 |
|  | Channel MLP | Linear 1 | 0.3211 | 0.3551 | 0.3477 | 0.3503 | 0.3436 |
|  |  | Linear 2 | 0.0247 | 0.0223 | 0.0205 | 0.0194 | 0.0217 |
|  |  | RepConv-1/2/3 | 0.3072 | 0.3335 | 0.3286 | 0.3310 | 0.3251 |
|  |  | $Q_S$ | 0.1468 | 0.1392 | 0.1392 | 0.1376 | 0.1407 |
|  |  | $K_S$ | 0.0373 | 0.0437 | 0.0442 | 0.0449 | 0.0426 |
|  | SDSA | $V_S$ | 0.0935 | 0.1255 | 0.1331 | 0.1333 | 0.1213 |
| Block6 |  | $K_S^T V_S$ | 0.3380 | 0.4449 | 0.4569 | 0.4667 | 0.4266 |
|  |  | $Q_S(K_S^T V_S)$ | 0.8073 | 0.8623 | 0.8706 | 0.8725 | 0.8532 |
|  |  | RepConv-4 | 0.2862 | 0.4085 | 0.4315 | 0.4352 | 0.3903 |
|  | Channel MLP | Linear 1 | 0.3084 | 0.3267 | 0.3192 | 0.3230 | 0.3193 |
|  |  | Linear 2 | 0.0241 | 0.0218 | 0.0202 | 0.0194 | 0.0214 |
|  | Downsampling | Conv | 0.2456 | 0.2487 | 0.2414 | 0.2438 | 0.2449 |
|  |  | RepConv-1/2/3 | 0.1662 | 0.3402 | 0.3052 | 0.3280 | 0.2849 |
|  |  | $Q_S$ | 0.2044 | 0.1330 | 0.1202 | 0.1096 | 0.1418 |
|  |  | $K_S$ | 0.0221 | 0.0259 | 0.0214 | 0.0205 | 0.0225 |
|  | SDSA | $V_S$ | 0.0870 | 0.1556 | 0.1438 | 0.1443 | 0.1327 |
| Block1 |  | $K_S^T V_S$ | 0.1782 | 0.2832 | 0.2455 | 0.2412 | 0.2370 |
|  |  | $Q_S(K_S^T V_S)$ | 0.6046 | 0.6607 | 0.5710 | 0.5285 | 0.5912 |
|  |  | RepConv-4 | 0.2379 | 0.2635 | 0.1852 | 0.1592 | 0.2115 |
|  | Channel MLP | Linear 1 | 0.2332 | 0.3966 | 0.3615 | 0.3859 | 0.3443 |
| stage4 |  | Linear 2 | 0.0262 | 0.0252 | 0.0192 | 0.0171 | 0.0219 |
|  |  | RepConv-1/2/3 | 0.3053 | 0.4001 | 0.3907 | 0.4018 | 0.3745 |
|  |  | $Q_S$ | 0.1389 | 0.1245 | 0.1176 | 0.1108 | 0.1230 |
|  |  | $K_S$ | 0.0227 | 0.0231 | 0.0224 | 0.0218 | 0.0225 |
|  | SDSA | $V_S$ | 0.0764 | 0.1038 | 0.1051 | 0.1048 | 0.0975 |
| Block2 |  | $K_S^T V_S$ | 0.1600 | 0.1968 | 0.1985 | 0.1979 | 0.1883 |
|  |  | $Q_S(K_S^T V_S)$ | 0.5439 | 0.5558 | 0.5348 | 0.5079 | 0.5356 |
|  |  | RepConv-4 | 0.1718 | 0.1697 | 0.1578 | 0.1384 | 0.1594 |
|  | Channel MLP | Linear 1 | 0.3000 | 0.3811 | 0.3768 | 0.3913 | 0.3623 |
|  |  | Linear 2 | 0.0030 | 0.0035 | 0.0032 | 0.0029 | 0.0032 |
|  | Head | Linear | 0.4061 | 0.4205 | 0.4323 | 0.4545 | 0.4283 |

