# OpenReview forum: "Spike-driven Transformer V2: Meta Spiking Neural Network Architecture Inspiring the Design of Next-generation Neuromorphic Chips"
_ICLR.cc/2024/Conference — ICLR 2024 poster_

### Official Review · Reviewer_1cdx · 2023-10-13

**Soundness:** 2 fair
**Presentation:** 3 good
**Contribution:** 2 fair
**Rating:** 5
**Confidence:** 4

**Summary:**

This work proposes a transformer-based spiking neural network, Meta-SpikeFormer, that can achieve 1) low power; 2) versatility; and 3) high accuracy by a meta architecture. This work explore the impact of structure, spike-driven self-attention, and skip connection on the performance of Meta-SpikeFormer. On ImageNet-1K, Meta-SpikeFormer achieves 80.0% top-1 accuracy (55M), surpassing the current state-of-the-art (SOTA) SNN baselines (66M) by 3.7%.

**Strengths:**

+ This work proposes a transformer-based spiking architecture consisting of RepConv2, SDSA, and ChannelMLP.
+ This work compares itself against prior SNN-based designs.

**Weaknesses:**

- The motivation of this work is weak. Why is SNN important?
- A metric of the result is not clear. How is the "power" value computed?

**Questions:**

1. Why is the SNN architecture interesting and important?
"On ImageNet-1K, Meta-SpikeFormer achieves 80.0% top-1 accuracy (55M)" is not the state-of-the-art. There are many prior works which can outperform this result. see: https://paperswithcode.com/sota/image-classification-on-imagenet

For example, tinyVIT https://www.ecva.net/papers/eccv_2022/papers_ECCV/papers/136810068.pdf
tinyVIT can achieve top-1 accuracy 80\% on imagenet-1k by only 5M parameters.

Another example is, efficientNet B3: https://proceedings.mlr.press/v97/tan19a/tan19a.pdf
efficientNet B3 can obtain top-1 accuracy 81.1\% on imagenet-1k by only 12M parameters.

---

> ### Author Response · Authors · 2023-11-20
> **Why is SNN important?**
>
> We sincerely thank you for your insightful feedback. Your question is fundamental and profound. It is something that every researcher in the SNN field should seriously think about. We'd be happy to discuss this with you. Here, we give some of our thoughts. Let's start with two examples.
>
> *Example 1.* The emergence of Large Language Models (LLMs) heralds the beginning of a new era of AI, however, one of the biggest concerns is that current LLMs consume a vast amount of energy. Compared with the one-time consumption in the training process, **the inference process can be repeated over and over again by different users and thus represents iterative consumption that uses much more energy**. If a new technology can reduce the energy cost of inference by 10 to 100 times, the cost of using AI will be significantly reduced. For commercial companies, cost reduction means increased competitiveness.
>
> *Example 2.* **The human brain performs much more complex tasks than LLMs, comprises a larger network (with 86 billion neurons and trillions of synapses), but only consumes 20W of power**. Thus, brain-inspired machine learning and computing are promising approaches to solving the energy problem. Neuromorphic computing exploits SNNs on neuromorphic chips and is a typical brain-inspired computing paradigm. **Spiking neurons simulate biological neurons' spatio-temporal dynamics and spike communication, which is biologically reasonable. Therefore, SNNs are important and attractive, which can be reflected in *at least* the following two aspects:**
>
> - **Efficiency.** SNNs have obvious power advantages over ANNs.** Its energy-efficient has been verified on small-scale models [1,2,3]and some edge computing scenarios [4,5,6]. In [1], SNN demonstrated $10^3$ to $10^4$ times energy efficiency compared to ANN on small-scale tasks. Another example is, a recent edge computing device called Speck integrates an SNN-enabled neuromorphic chip with a DVS camera. Its peak power on gesture recognition task is 0.4~15mW (https://www.synsense.ai). Moreover, some other general and well-known neuromorphic chips, such as TruthNorth [7], Loihi [8], Tianjic [9], consume only a few hundred mW of energy. If SNNs can capitalize on their low-power in large-scale models, they will greatly reduce the cost of using AI.
>
> - **Effectiveness.** Neuroscience has long been an essential driver of progress in AI, but current AI is very different from how the human brain works [10]. A classic example is that of biological versus artificial neural networks. Current ANNs use low-complexity artificial neurons and super-sized networks to achieve machine intelligence. However, the complexity of biological neurons is much higher than artificial neurons [11]. Another possible way is a combined approach of high-complexity spiking neurons and networks of a certain size. Thus, brain-like SNNs can offer more possibilities for realizing machine intelligence.
>
> Overall, as a model of brain-inspired computing model, SNN is expected to be an alternative to traditional AI [12].
>
> ---
> [1] Yin, Bojian, Federico Corradi, et al. "Accurate and efficient time-domain classification with adaptive spiking recurrent neural networks." In *Nature Machine Intelligence*, 2021.
>
> [2] Rao, Arjun, Philipp Plank, et al. "A long short-term memory for AI applications in spike-based neuromorphic hardware." In *Nature Machine Intelligence*, 2022.
>
> [3] Shaban, Ahmed, Sai Sukruth Bezugam, et al. "An adaptive threshold neuron for recurrent spiking neural networks with nanodevice hardware implementation." In *Nature Communications*, 2021.
>
> [4] Gallego, Guillermo, Tobi Delbrück, et al., "Event-Based Vision: A Survey," in *IEEE T-PAMI*, 2022.
>
> [5] Zhou, Yue, Jiawei Fu, et al. "Computational event-driven vision sensors for in-sensor spiking neural networks." In *Nature Electronics*, 2023.
>
> [6] Ma, Songchen, Jing Pei, et al. "Neuromorphic computing chip with spatiotemporal elasticity for multi-intelligent-tasking robots." In *Science Robotics*, 2022.
>
> [7] Merolla, Paul A., et al. "A million spiking-neuron integrated circuit with a scalable communication network and interface." In *Science*, 2014.
>
> [8] Davies, Mike, Narayan Srinivasa, et al. "Loihi: A neuromorphic manycore processor with on-chip learning." In *IEEE Micro*, 2018.
>
> [9] Pei, Jing, Lei Deng, et al. "Towards artificial general intelligence with hybrid Tianjic chip architecture." In *Nature*, 2019.
>
> [10] Zador, Anthony, Sean Escola, et al. "Catalyzing next-generation artificial intelligence through neuroai." In *Nature communications*, 2023.
>
> [11] Herz, Andreas VM, et al. "Modeling single-neuron dynamics and computations: a balance of detail and abstraction." In *Science*, 2006.
>
> [12] Roy, Kaushik, Akhilesh Jaiswal, et al. "Towards spike-based machine intelligence with neuromorphic computing." In *Nature*, 2019.

---

> ### Author Response · Authors · 2023-11-20
> **Why is the SNN architecture interesting and important?**
>
> > **Q:** Why is the SNN architecture important?
>
> **A:** As a **cutting-edge cross-direction based on neuroscience and computer science**, the ambition of SNNs is to become a low-power alternative to traditional machine intelligence [1]. This makes neuromorphic computing appear as a systematic project, covering algorithms, hardware, etc [2]. **Therefore, SNN architecture should not be understood separately from neuromorphic systems.**
>
> - **Advantages of Neuromorphic Computing.** Neuromorphic chip is non-von Neumann architecture hardware whose structure and function are inspired by brains. Some unique fundamental operational nature [3], including highly parallel operation, collocated processing and memory, inherent scalability, and spike(event)-driven computation, stem from their choice to incorporate spiking neurons and synapses as primary computational units. **Thus, the importance of SNN architectures is first and foremost reflected in the unique computational nature of SNN-enabled neuromorphic chips.**
>
> - **Why can't current SNNs go mainstream?** We've covered the importance of SNNs in our last response. However, the reality is that ANNs are the mainstream of machine intelligence, SNNs are not. The reason is that the advantages of SNNs are hard to be shown. In terms of energy consumption, SNNs can only realistically demonstrate their advantages when deployed on neuromorphic chips, which is a big job. In terms of performance, the gap between SNNs and ANNs is obvious.
>
> - **How to close the gap between SNNs and ANNs?** The main dilemmas currently faced by the SNN domain are: (1) **No standard baseline in SNNs**, which means that researchers can't combine their efforts to move the field forward quickly. (2) **Prior SNN works failed in other tasks** due to low accuracy and no standard architecture. **In this work, we provide a standard meta baseline for the SNN domain, and the proposed model is highly accurate and can be used on multiple tasks in a unified way.**
>
> **Therefore, the SNN architecture can not only take advantage of neuromorphic computing chips, but an excellent SNN architecture can also narrow the performance gap between SNNs and ANNs.**
>
> > **Q:** Why is the SNN architecture interesting?
>
> **A:** Because SNN architecture mimics the structure and function of the human brain more closely than traditional ANNs.
>
> We take the attention mechanism as an example. An important function of the human brain is to dynamically allocate responses according to task difficulty, which is often referred to as attention. Salient stimuli are typically given more attention, mainly reflected in the more intense spiking firing of brain regions or neurons related to the stimulus.
>
> In [5], an additional attention module was incorporated into spiking CNN to suppress noisy features and focus on important features. Subsequently, the spike firing of SNN was significantly reduced, and the task performance was improved. Thus, **the combination of attention and SNN can achieve higher performance with less energy cost, which is consistent with the function of the attention mechanism in the human brain [6]**.
>
> In this work, we explore spiking Transformer and spike-driven self-attention operators. We also found some interesting points, e.g., the interactions between $Q, K,$ and $V$ are sparse and linear. We expect future SNN architectures to be as computationally sparse and effective as the human brain.
>
>
> ---
> [1] Roy, Kaushik, et al. "Towards spike-based machine intelligence with neuromorphic computing." In *Nature*, 2019.
>
> [2] Mehonic, Adnan, et al. "Brain-inspired computing needs a master plan." In *Nature*, 2022.
>
> [3] Schuman, et al. "Opportunities for neuromorphic computing algorithms and applications." In *Nature Computational Science*, 2022.
>
> [4] Su, Qiaoyi, et al. "Deep directly-trained spiking neural networks for object detection." In *ICCV*, 2023.
>
> [5] Yao, Man, et al. "Attention spiking neural networks." In *IEEE T-PAMI*, 2023.
>
> [6] Briggs, Farran, et al. "Attention enhances synaptic efficacy and the signal-to-noise ratio in neural circuits." In *Nature*, 2013.

---

> ### Author Response · Authors · 2023-11-20
> **Other Questions**
>
> > **Q:** A metric of the result is not clear. How is the "power" value computed?
>
> **A: The concept of "Power" is designed to facilitate the comparison of energy between ANNs and SNNs, which is a common practice in SNNs.** FLOPs are commonly used in ANNs to evaluate the cost of a model, where one FLOP can be thought of as a single Multiply-and-ACcumulate (MAC) [1]. In ANNs, almost all FLOPs are MACs. In contrast, there are only sparse ACcumulate (AC) in SNNs because binary spike signals are used for communication. The energy cost of ANNs is FLOPs times $E_{MAC}$. The energy cost of SNNs is FLOPs times $E_{AC}$ times spike firing rate. $E_{MAC} = 4.6pJ$ and $E_{AC} = 0.9pJ$ are the energy of a MAC and an AC, respectively, in 45nm technology [2]. We provide detailed energy evaluation methods in **Appendix~B**. The spike firing rate of the proposed network is in Table~11.
>
> **Note, the above analysis is a theoretical evaluation and ignores details such as hardware architecture or data caching**. The efficiency advantages of neuromorphic computing may be even greater when SNNs are deployed on neuromorphic chips. For example, due to the spike-driven nature, neuromorphic chips can be designed in asynchronous mode without a global clock. When there is no spike input, the entire chip has almost no static power, such as Speck designed by SynSense (https://www.synsense.ai). Generally, the power of neuromorphic computing is positively related to the number of ACs brought by the spikes [3].
>
> > **Q:** On ImageNet-1K, Meta-SpikeFormer achieves 80.0% top-1 accuracy (55M)" is not the state-of-the-art. There are many prior works which can outperform this result. see: paperwithcode, tinyVIT and efficientNet B3.
>
> **A:** Yes, this is not the state-of-the-art result on the ImageNet-1K dataset. Note, **we claim that our work is the SOTA result in the SNN domain on ImageNet-1K**. Although the current Meta-SpikeFormer does not show better accuracy than the fully optimized TinyViT, this field is developing rapidly. **It took the SNN field less than a year to improve the performance of the prior spiking Transformer [4] in ICLR 2023 on ImageNet-1K by 5.2%.** Our model is designed strictly according to the spike-driven. It can be advantageous if it can inspire the design of next-generation neuromorphic chips and be fully optimized at the algorithm level.
>
> After the emergence of ViT [5], it took three years of relentless work on the SNN before the first true spiking Transformer appeared. We provide a standard meta baseline for the SNN domain, and the proposed model is highly accurate and can be used on multiple tasks in a unified way. **It will undoubtedly advance the domain of neuromorphic computing. The SNN domain can narrow the gap between ANN and SNN from 30 years to 2-3 years.** The back-propagation algorithm was proposed in 1986 [6]. SNN was called the third-generation neural network in 1997 [7], but it was in 2018-2019 that the SNN field solved the basic training problem [8], i.e., spatio-temporal BP became the default direct training algorithm for SNN. **We are very confident that our model and codebase will become a landmark work in SNN and push the field of neuromorphic computing further at the algorithmic and hardware levels. It is only a matter of time before the gap between SNNs and ANNs will soon be further minimized based on the proposed meta baseline.**
>
> In Table 5, we perform full ablation experiments and reveal the multiple tradeoffs between performance, power, and parameters present in the Meta-SpikeFormer. Subsequent researchers can build on our work and optimize it in depth for their own needs, just like tinyVIT and efficientNet B3 are optimizations of prior ViT and CNN baselines.
>
> ---
> [1] Molchanov, et al. "Pruning convolutional neural networks for resource efficient inference." In *ICLR*, 2017.
>
> [2] Horowitz, Mark. "1.1 computing's energy problem (and what we can do about it)." In *IEEE ISSCC*, 2014.
>
> [3] Pei, Jing, et al. "Towards artificial general intelligence with hybrid Tianjic chip architecture." In *Nature*, 2019.
>
> [3] Eshraghian, et al. "Training spiking neural networks using lessons from deep learning." In *Proceedings of the IEEE*, 2023.
>
> [4] Zhou, Zhaokun, et al. "Spikformer: When spiking neural network meets transformer." In ICLR, 2023.
>
> [5] Dosovitskiy, et al. "An image is worth 16x16 words: Transformers for image recognition at scale." In ICLR, 2021.
>
> [6] Rumelhart, et al. "Learning representations by back-propagating errors." In *Nature*, 1986.
>
> [7] Maass, Wolfgang. "Networks of spiking neurons: the third generation of neural network models." In *Neural networks*, 1997.
>
> [8] Neftci, Emre O.,  et al. "Surrogate gradient learning in spiking neural networks: Bringing the power of gradient-based optimization to spiking neural networks." In *IEEE Signal Processing Magazine*, 2019.

---

### Official Review · Reviewer_JmyY · 2023-10-31

**Soundness:** 4 excellent
**Presentation:** 4 excellent
**Contribution:** 3 good
**Rating:** 6
**Confidence:** 4

**Summary:**

The paper extends the previous Spike-driven Transformer into a meta-structure with new macro-level conv and new self-attention SNN block designs, achieving SOTA accuracies on four types of vision tasks with controlled model parameters.

**Strengths:**

- Clear paper structure and easy-to-follow presentation.
- Compared to the previous Spiken-driven Transformer, the proposed new one achieves the best accuracy.
- Extensively evaluated on four vision tasks: image classification, event-based action recognition,  currently the largest event-based human activity recognition dataset), object detection, and semantic segmentation.
- Novel new self-attention SNN implementation that contributes to accuracy gain.

**Weaknesses:**

* Only focus on ViT. Although the paper claims a general spike-driven Transformer, it only discusses vision tasks and elaborates on a Transformer design for vision tasks with a two-stage Conv Block. It would be more beneficial to add a discussion on how to extend to language tasks.

**Questions:**

* Can the authors comment on how to extend the proposed techniques to Transformer for language tasks?
* Can the authors provide the training costs of the Spiking Transformer with vanilla Transformer models?

---

> ### Author Response · Authors · 2023-11-20
>
> Thank you for your insightful feedback. Your questions are crucial and something we have been thinking about.
>
> > **Weakness & Q1:** Only focus on ViT. Although the paper claims a general spike-driven Transformer, it only discusses vision tasks and elaborates on a Transformer design for vision tasks with a two-stage Conv Block. It would be more beneficial to add a discussion on how to extend to language tasks. Can the authors comment on how to extend the proposed techniques to Transformer for language tasks?
>
> **Q:** We sincerely appreciate your advice. Not testing the generality of our architecture on language tasks is indeed a limitation of this paper. We will add this point in the Limitations Section (first paragraph of the supplementary material).
>
> Regarding the language task, we are actually currently working in this direction, since Large Language Model (LLM) is one of the hottest topics at the moment, SNN domain should not miss it. However, the challenges in language tasks are different from vision tasks. If the goal is to achieve spiking LLM, we believe that challenges exist in several aspects:
>
> - **Modeling of long-term dependencies in the temporal dimension.** There is already work in this area [1], but it remains to be verified whether it can still be effective on longer input sequences and larger networks.
>
> - **Parallel spiking neuron design.** If SNNs cannot be trained in parallel, the training cost required is unaffordable as the length of the input sequences continues to increase. A recent work at NeurIPS 2023 presented the first parallel spiking neuron [2]. However, it forcibly removed the reset function, which affects the spatio-temporal dynamics as well as the performance of the spiking neuron.
>
> - **Network architecture.** Transformer-based models are widely used in language tasks. Therefore, it is desirable to adopt a similar approach in the SNN domain.
>
> - **Pre-training techniques.** The vision spiking Transformer usually uses supervised training. Whereas the training of large models in language tasks usually exploits self-supervised pre-training.
>
> Overall, the techniques presented in this paper can provide inspiration for SNNs to handle language tasks, at least in **Modeling of long-term dependencies in the temporal dimension** and **Network architecture**. Since we investigate the meta Transformer-based SNN architecture, from the perspective of meta-design, it can be easily noticed that the architecture of Meta-SpikeFormer is similar to those in many LLMs [3]. For example, a common architecture in NLP is temporal (token) mixer + channel mixer, which is essentially the same as what is presented in this paper in Fig. 1. Moreover, this paper proposes three Spike-Driven Self-Attention (SDSA) operators that can provide support for long-term dependence modeling.
>
> > **Q2:** Can the authors provide the training costs of the Spiking Transformer with vanilla Transformer models?
>
> **A:** Using two NVIDIA A100 (40G) GPUs as an example, the training speeds for these two models are as follows.
>
> - Meta-SpikeFormer (This work): 55M, BatchSize=100*2, 52.07 min/epoch
> - ViT-based [4]: 87M, BatchSize=256*2, 17.08 min/epoch
>
> Overall, SNNs’ training is more consuming, which is caused by the multiple timesteps of SNNs. This is a long-standing problem in the field of SNN.
>
> ---
> [1] Rao, Arjun, Philipp Plank, et al. "A long short-term memory for AI applications in spike-based neuromorphic hardware." In *Nature Machine Intelligence*, 2022.
>
> [2] Fang, Wei, Zhaofei Yu, et al. "Parallel Spiking Neurons with High Efficiency and Long-term Dependencies Learning Ability." In NeurIPS, 2023.
>
> [3] https://github.com/RUCAIBox/LLMSurvey
>
> [4] Dosovitskiy, Alexey, Lucas Beyer, et al. "An image is worth 16x16 words: Transformers for image recognition at scale." In *ICLR*, 2021.

---

> > ### Comment · Reviewer_JmyY · 2023-11-22
> >
> > Thanks for the detailed response. Please include limitations in the manuscript.

---

> > > ### Author Response · Authors · 2023-11-23
> > >
> > > Thanks for your kind reminder, we have updated the Limitation Section (blue words in first paragraph of the supplementary material) to include some discussion about using Meta-SpikeFormer for language tasks, and the training cost caused by multiple timesteps.

---

### Official Review · Reviewer_Aepg · 2023-10-31

**Soundness:** 3 good
**Presentation:** 3 good
**Contribution:** 2 fair
**Rating:** 6
**Confidence:** 3

**Summary:**

The authors describe a new spiking transformer architecture and
investigate the performance on a number of benchmarks, such as image
classification (ImageNet-1k), object detection and activity
recognition. They based their architecture mainly on the previously published
spike-driven transformer (Yao et al. 2023), however, more initial conv layers are
added, the attention mechanism is changed somewhat, as well as some
other previously published aspects are incorporated (e.g. repconv, sepconv). The
model improves the SNN result for ImageNet compared to earlier works,
and shows good performances on the other benchmarks as well.

**Strengths:**

In general, it is a well written paper discussing an improved model
architecture and showing its performance. While performance is indeed
improved the architecture seems very much in spirit of Yao et al. with
some tweaks (e.g. more CNN layers) and a changed attention (matrix
product instead of hadamard, which however, has worse
complexity).   Here, it is shown additionally that the
model performs well on other tasks as well, which is,
however, generally known for a transformer-like architectures and
thus not very surprising. The comparison and ablations with different
attention mechanisms and short-cut structures is interesting.

**Weaknesses:**

While the study is well done, the contributions are
rather incremental as the bulk of the model architecture was
presented in earlier papers.

**Questions:**

- The main argument and novelty of the paper seems to be the "meta"
aspect of the architecture. I am not following the argument made
here. What exactly is meant with "meta" here? Its broad versatility for
different datasets?

- While the accuracy on ImageNet-1k is improved compared to Yao et
al. (79.7\% versus 76.3\%), the energy consumption is almost 10x the
number (52.5 versus 6.1 mJ, according to Table 1). Maybe this is due
to the more complex attention algorithm (although in the ablation
table 5 only a moderate energy reduction from SDSA-3 -> SDSA-1 is
mentioned)?  However, in section 4.1 the power numbers written look
very comparable between the models (11.9 versus 10.2mJ) since one is
apparently stated for T=1 and the other for T=4. Should one not
compare T=4 with T=4? Would be also helpful to discuss the 10x power
increase and the dependence on T.

---

> ### Author Response · Authors · 2023-11-20
> **About the contribution**
>
> Thanks for your insightful feedback and your time in reading our paper.
>
> > **Weaknesses:** While the study is well done, the contributions are rather incremental as the bulk of the model architecture was presented in earlier papers.
>
> **A:** We would like to start by introducing the history of the spiking Transformer.
>
> Since ViT [1] was proposed in 2020, there has been a proliferation of architectural designs for vision Transformers. Incorporating the effectiveness of Transformer with the low power of SNNs is a natural and exciting idea. There has been some work in this direction, but most work simply replaces a part of the neurons in the Transformer with spiking neurons. It wasn't until that SpikFormer [2] spiked almost the entire network. Further, spike-driven Transformer [3] incorporates spike-driven nature into Transformer. Their contributions are:
>
> - **SpikFormer [2] in ICLR 2023.** SpikFormer designed self-attention interactions between spike-form $Q$, $K$, $V$ for the first time, which was an important breakthrough. SpikFormer 's residual connection makes it possible to pass multi-spike (integer) signals between spiking neurons. Thus, SpikFormer is actually an integer-driven Transformer.
>
> - **Spike-driven Transformer [3] in NeurIPS 2023.** Spike-driven Transformer advanced SpikFormer by redesigning Spike-Driven Self-Attention (SDSA) and shortcut. There is only sparse addition in Spike-driven Transformer.
>
> After the emergence of ViT, it took three years of relentless work on the SNN domain before the first true spiking Transformer appeared. **Integrating the benefits of Transformer in SNNs is non-trivial**. The main dilemmas currently faced are:
>
> - **No standard baseline in SNNs**, which is fatal to the development of an ambitious and cutting-edge field. This means that researchers can't combine their efforts to move the field forward quickly.
>
> - **Gap between SNN and ANN.** Prior SNN works failed in other tasks due to low accuracy and no standard architecture.
>
> In this work, we provide a standard meta baseline for the SNN domain, and the proposed model is highly accurate and can be used on multiple tasks in a unified way. **This work will undoubtedly advance the domain of neuromorphic computing. The SNN domain can narrow the gap between ANN and SNN from 30 years to 2-3 years.** The back-propagation algorithm was proposed in 1986 [4]. SNN was called the third-generation neural network in 1997 [5], but it was in 2018-2019 that the SNN field solved the basic training problem [6,7], i.e., spatio-temporal BP became the default direct training algorithm for SNN. This work now improved the performance of the SOTA SNN in NeurIPS 2023 on ImageNet by 3.7\% through the architectural design, and achieved good results on dense prediction tasks in a unified manner for the first time. It is less than a year after SpikFormer [2] in ICLR 2023 was proposed (our model is 5.2\% higher than SpikFormer on ImageNet-1K).
>
> **We are very confident that our model and codebase will become a landmark work in the SNN domain and push the field of neuromorphic computing further at the algorithmic and hardware levels.** We think that it is only a matter of time before the gap between SNNs and ANNs will soon be further minimized based on the meta baseline in this paper.
>
> ---
> [1] Dosovitskiy, Alexey, Lucas Beyer, et al. "An image is worth 16x16 words: Transformers for image recognition at scale." In *ICLR*, 2021.
>
> [2] Zhou, Zhaokun, Yuesheng Zhu, et al. "Spikformer: When spiking neural network meets transformer." In *ICLR*, 2023.
>
> [3] Yao, Man, Jiakui Hu, et al. "Spike-driven transformer." In *NeurIPS*, 2023.
>
> [4] Rumelhart, David E., Geoffrey E. Hinton, and Ronald J. Williams. "Learning representations by back-propagating errors." In *Nature*, 1986.
>
> [5] Maass, Wolfgang. "Networks of spiking neurons: the third generation of neural network models." In *Neural networks*, 1997.
>
> [6] Wu, Yujie, Lei Deng, et al. "Spatio-temporal backpropagation for training high-performance spiking neural networks." In *Frontiers in neuroscience*, 2018.
>
> [7] Neftci, Emre O., Hesham Mostafa, et al. "Surrogate gradient learning in spiking neural networks: Bringing the power of gradient-based optimization to spiking neural networks." In *IEEE Signal Processing Magazine*, 2019.

---

> > ### Author Response · Authors · 2023-11-20
> > **About the Questions**
> >
> > > **Q1:** The main argument and novelty of the paper seems to be the "meta" aspect of the architecture. I am not following the argument made here. What exactly is meant by "meta" here? Its broad versatility for different datasets?
> >
> > **A: "Meta" refers to the meta-architecture design of Transformer-based SNN. The goal of meta design in this work is to provide a powerful baseline model for the SNN domain on a variety of tasks, and in doing so, advance the entire domain.**
> >
> > Technically, the design of the meta-architecture in this work includes the following three aspects:
> >
> > - **Network structure.** The hybrid design of Conv+ViT is beneficial to the accuracy and versatility of SNNs. Given our meta Conv/Transformer-based blocks, **subsequent researchers can perform targeted optimization of the design details inside the meta blocks**.
> >
> > - **Shortcut.** We discussed several shortcuts in SNN. We point out that Spike-Element-Wise-based spiking Transformer is an "integer-driven" rather than a "spike (binary)-driven" SNN. The membrane potential shortcut is fine with this problem.
> >
> > - **Spike-Driven Self-Attention (SDSA).** SDSA is the core design of long-distance dependency modeling. But this is a design that current neuromorphic chips lack. **We propose three new SDSA operators**. We hope this will inspire the SNN domain and expect subsequent researchers to design more powerful SDSAs.
> >
> > > **Q2:** While the accuracy on ImageNet-1k is improved compared to Yao et al. (79.7% versus 76.3%), the energy consumption is almost 10x the number (52.5 versus 6.1 mJ, according to Table 1). Maybe this is due to the more complex attention algorithm (although in the ablation table 5 only a moderate energy reduction from SDSA-3 -> SDSA-1 is mentioned)?
> >
> > **A:** The proposed Meta-SpikeFormer does have a significant improvement in energy cost compared to Yao et al [1]. The main reason for this is that we focus mainly on the design of the meta-architecture to explore more possibilities, which may introduce some redundancy in the design. **We perform full ablation experiments in Table 5 and reveal the multiple tradeoffs between performance, power, and parameters present in the Meta-SpikeFormer architecture**.
> >
> > We analyzed the energy cost of Meta-SpikeFormer and found that it is the Conv-based SNN block that brings a significant increase in power, not the SDSA. As shown in Table 5, if SDSA-3 -> SDSA-1, Power: 7.8->7.6; if remove SepConv part in Conv-based SNN block, Power: 7.8-> 5.5. **Since we provide a meta baseline for the SNN domain, subsequent researchers can build on our work and optimize it in depth for their own needs.**
> >
> > > **Q3:** However, in section 4.1 the power numbers written look very comparable between the models (11.9 versus 10.2mJ) since one is apparently stated for T=1 and the other for T=4. Should one not compare T=4 with T=4? Would be also helpful to discuss the 10x power increase and the dependence on T.
> >
> > **A:** Honestly, it is a writing trick that we use the T=1 power to compare with the previous T=4 power. But it should also be noted that our accuracy on T=1 is higher than the previous T=4 accuracy.
> >
> > Moreover, you raise a very profound point. The timestep issue is a persistent problem in the SNN domain for a long time. When SNNs handle image classification tasks, the same image will be input repeatedly at all timesteps. As you can observe in Table 1, the accuracy of T=4 (repeated input 4 times) will indeed be higher than that of T=1 (about 1% to 2% improvement), which is a common processing method in the SNN domain. However, multiple timesteps significantly increase training time, hardware requirements, power, etc. Thus, in image classification task, multiple timesteps are actually a no-brainer option to ensure accuracy.
> >
> > ---
> > [1] Yao, Man, Jiakui Hu, et al. "Spike-driven transformer." In *NeurIPS*, 2023.

---

> > > ### Comment · Reviewer_Aepg · 2023-11-20
> > >
> > > Many thanks for the detailed responses.

---

### Author Response · Authors · 2023-11-23
**General Response to ACs and Reviewers**

Dear ACs and Reviewers,

We sincerely thank for the valuable comments. **We confidently believe that this work is worthy of being accepted by ICLR, because it advances the neuromorphic computing domain in a tangible way, without bells and whistles.** We explain the reasons as follows:

- **SNN is important.** As a cutting-edge cross-direction based on neuroscience and computer science, neuromorphic computing (SNN + neuromorphic chip) has many appealing advantages (please refer to the response to Reviewer 1cdx for more details), such as bio-plausibility and low power consumption.

- **A Major concern in SNN** is that there is a large performance gap between SNNs and ANNs. The gap is narrowing, but not enough. The SNN domain can narrow the gap between ANNs and SNNs from 30 years to 2-3 years. The BP was proposed in 1986. It was not until 2018-2019 that the SNN field solved the basic training challenge. ViT was presented at ICLR 2021, but until NeurIPS 2023 [1], it was the first spike-driven Transformer.

Now, we have pushed the field further, **our contributions:**

- **A standard meta baseline.** The meta architecture design enables subsequent researchers to quickly advance the field based on our work.

- **Versatility.** This work is **the first** direct training SNN backbone to handle image classification, object detection, and semantic segmentation concurrently.

- **Performance.** This work improves the state-of-the-art results of the SNN domain **on ImageNet classification by 3.7%**, state-of-the-art results on HAR-DVS (currently the largest event-based human activity recognition dataset) and COCO, first semantic segmentation baseline on ADE20K.

**All three Reviewers did not question our technical details, experiments, and results in any way.** The main concern of the Reviewers was the importance of SNNs or the difference in contribution between this work and the previous Spike-driven Transformer [1]. We named Meta-SpikeFormer as Spike-driven Transformer V2 (following the name of [1]) because serialized works can attract more attention. Rapid advances in neuromorphic computing require a concerted effort by researchers in the field. It is only a matter of time before the gap between SNNs and ANNs will soon be further minimized based on the meta baseline in this paper.

We really hope that the ACs and the Reviewers can give more support to this new research direction.

Best regards

Authors

---
[1] Yao, Man, et al. "Spike-driven transformer." In NeurIPS, 2023.

---

### Meta-Review · Area_Chair_i88V · 2023-12-18

**Metareview:**

This paper proposes "Meta-SpikeFormer" a new Transformer-based spiking neural network architecture. On ImageNet-1K, Meta-SpikeFormer achieves state-of-the-art results in the SNN domain. Authors show that their model consumes less power and it also versatile.

The only negative comment about the paper is by 1cdx who rejects the paper simply because the performance in ImageNet is not better than normal neural network performance. I think it is ok since this paper aims to close the gap between ANNs and SNNs.

This is an important research direction and not a lot of researchers are working on it. I recommend an acceptance.

**Justification For Why Not Higher Score:**

This is solid incremental work hence the poster.

**Justification For Why Not Lower Score:**

There is no reason to reject the paper.

---

### Decision · Program_Chairs · 2024-01-16

Accept (poster)